# Amortized Mixing Coupling Processes for Clustering

**Huafeng Liu**[1,2], **Liping Jing**[1]

[1]Beijing Key Lab of Traffic Data Analysis and Mining, Beijing Jiaotong University, Beijing, China
[2]The Department of Mathematics, The University of HongKong, Hong Kong SAR, China
`huafeng.liu@outlook.com, lpjing@bjtu.edu.cn`

## Abstract

Considering the ever-increasing scale of data, which may contain tens of thousands of data points or complicated latent structures, the issue of scalability and algorithmic efficiency becomes of vital importance for clustering. In this paper, we propose cluster-wise amortized mixing coupling processes (AMCP), which is able to achieve efficient amortized clustering in a well-defined non-parametric Bayesian posterior. Specifically, AMCP learns clusters sequentially with the aid of the proposed intra-cluster mixing (IntraCM) and inter-cluster coupling (InterCC) strategies, which investigate the relationship between data points and reference distribution in a linear optimal transport mixing view, and coupling the unassigned set and assigned set to generate new cluster. IntraCM and InterCC avoid pairwise calculation of distances between clusters and reduce the computational complexity from quadratic to linear in the current number of clusters. Furthermore, cluster-wise sequential process is able to improve the quick adaptation ability for the next cluster generation. In this case, AMCP simultaneously learns what makes a cluster, how to group data points into clusters, and how to adaptively control the number of clusters. To illustrate the superiority of the proposed method, we perform experiments on both synthetic data and real-world data in terms of clustering performance and computational efficiency.

## 1 Introduction

Clustering aims to group samples into a certain number of clusters, such that similar samples belong to the same cluster, while dissimilar ones to different clusters. Aside from its usefulness in many downstream tasks, clustering is an important but challenging topic which has seen applications in various settings including social-media recommendation [32], customer partitioning [5], discovering social networks [7] and partitioning protein-protein interaction networks [3]. Among various clustering algorithms [37], probabilistic generative clustering models [39] have been widely concerned because of its flexibility and interpretability.

Probabilistic generative clustering models (as known as, mixture models) are a staple of statistical modeling in which a discrete latent variable is introduced for each observation, indicating its mixture component identity. These generative clustering models can be roughly divided into two categories: finite mixture model [6] and infinite mixture model [31]. In recent years, finite mixture models have been increasingly applied in unsupervised learning problems with the aid of deep neural networks. Instead of using an arbitrary prior for the latent variable, these methods adopted finite mixture prior, such as Gaussian mixture model (GMM) [14, 38]. A finite mixture model with a fixed number of clusters may fit the given dataset well, while it may be sub-optimal to use the same number of clusters if more data comes under a slightly changed distribution [31]. It would be ideal if the clustering models can figure out the unknown number of clusters automatically. Alternatively, infinite mixture model is the application of nonparametric Bayesian techniques to mixture modeling, which allows for the automatic determination of an appropriate number of mixture components. The prior dis-

tribution can be specified in terms of a data point sequential process called Dirichlet process (e.g., Chinese restaurant process (CRP)[29]), where the number of clusters can arbitrarily grow to better accommodate data as needed [12]. However, the inference of nonparametric posterior largely rely on the sequential data point modeling (e.g., Markov Chain Monte Carlo (MCMC)), which is time-consuming, with convergence that is difficult to assess.

While above probabilistic clustering is generally approached as an unsupervised learning problem, the amortization power of non-linear transformation (e.g., neural networks) has recently spurred progress on supervised formulations, based on novel tractable objective functions for generic data and an arbitrary number of clusters [20, 21, 27]. In exchange for the additional cost of procuring labeled training data and training the model, amortized clustering offers several benefits on top of time efficiency at test time. Traditional unsupervised clustering, especially probabilistic clustering, defines specific likelihood and prior on the data geometry, even without any prior knowledge of the data, which are not needed with amortized clustering, since the model learns based on the statistics of the labeled training data [21]. Moreover, to achieve the best result (e.g., agreement with human labeling or with a well-defined posterior), unsupervised methods usually need parameter tuning and post-processing (e.g., cluster merging) [15], which are obviated with amortized inference by implicitly learning the definition of clusters underlying the training datasets. In a sense this shares a similar philosophy as Neural Processes [10, 9, 16], which learns from multiple datasets to learn a prior over functions in a meta learning manner.

In this paper, motivated by these prior work, we propose cluster-wise Amortized Mixing Coupling Processes (AMCP), which is able to provides an efficient clustering architecture by grouping data points in an optimal transport cluster-wise view. In each cluster-wise step, AMCP generates cluster sequentially and finds one cluster a time. In this case, two key strategies, Intra-Cluster Mixing (*IntraCM*) and Inter-Cluster Coupling (*InterCC*), are proposed to measure the relations between different data points and generate cluster. Specifically, *IntraCM* focus on investigating the relationship between data points and reference distribution in a linear optimal transport view [33], which avoid pairwise calculation of distances between clusters and reduce the computational complexity from quadratic to linear in the current number of clusters. Then, *InterCC* leverages new insights on defining dissimilarity between assigned data points and the unassigned data points based on the couplings obtained in *IntraCM*, which is able to select the dissimilar data points to form the next cluster and update existing clusters. The key contributions of our work are as follows: (1) *Efficiency*. The proposed AMCP not only inherent the efficient cluster-wise learning manner, but also reduce the quadratic cost of traditional similarity weight computation with the aid of optimal transport mixing. (2) *Novelty*. To the best of our knowledge, AMCP is the first to introduce optimal transport for amortized clustering and effectively correlate the relationship between optimal transport and mixture models. (3) *Adaptivity*. Non-parametric learning manners exist not only in model parameter learning but also in cluster generation, which allows unsupervised parameter learning and non-handcrafted intervention. (4) *Flexibility*. Different from existing amortized clustering methods without changing the already-generated clusters, AMCP is able to dynamically adjust the previous generated clusters to avoid the accumulated errors with the aid of mixing and coupling manners. (5) *Effectiveness*. Extensive experiments conducted on both synthetic and real-world datasets demonstrate that AMCP can cluster data effectively and efficiently.

## 2  Related Work

In this section, we briefly introduce some recent developments in several related topics, namely probabilistic clustering and amortized clustering.

**Probabilistic Clustering** Probabilistic clustering models (or equivalently, mixture models) are a staple of statistical modeling in which a discrete latent variable is introduced for each observation, indicating its mixture component identity. There is a growing interest in developing probabilistic clustering methods using parametric or nonparametric prior for complex data modeling [6, 31, 14, 38], and finite and infinite mixture models were developed. The main focus of finite model introduces finite mixture prior with a fixed number of cluster to clustering. In this case, Gaussian distribution or non Gaussian distributions [24, 23] are widely used to model data distribution [6]. Alternately, infinite mixture models [2] adopt non-parametric prior to control the number of clusters dynamically, such as Dirichlet process and its variants [29]. Recently, researchers focus on deep generative clustering by combining probabilistic clustering modeling and neural networks [14, 22, 27, 38, 21, 36, 4]. The

main focus of these methods is to learn a representation of input data amenable to clustering via deep neural networks.

**Amortized Clustering** Clustering is generally formulated as an unsupervised learning manner and while amortized clustering [27, 20, 21] investigates the amortized power steam from a predefined supervised formulation to achieve an arbitrary number of clusters in new task. Lee et al. [20, 21] adopt attention networks to output instead the cluster labels of each data point, for general mixture model, with an arbitrary number $K$ of clusters and a cost of $O(K)$ forward passes. Later, Pakman et al. [27] presented Neural Clustering Process (NCP), which sequentially computes the conditional probability of assigning the current data point to one of already constructed clusters or a new one. The most similar work to our methods is NCP [27] and CHiGac [25]. Both of them explain clustering from the perspective of generative models, which similar in spirit to the popular Gibbs sampling algorithm for Dirichlet process mixture models, but without positing particular priors on partitions. Besides, NCP and CHiGac rely heavily on the selecting of anchor points and the processing ordering and often exhibits unstable properties. Different from previous work, AMCP connects the relationship of mixture model and optimal transport to describe the cluster's summary statistics well, and seamlessly combines mixing and coupling to achieve efficient amortized clustering.

## 3 Method

### 3.1 Notations and Problem Formulation

Given $N$ data points $\mathcal{D} = \{\mathbf{x}_1, \mathbf{x}_2, \cdots, \mathbf{x}_N\}$, where $\mathbf{x}_i \in \mathbb{R}^d$ indicates feature representation for the $i$-th data point represented in $d$-dim latent space. Probabilistic models for clustering is usually presented by sequential sampling procedure to generate clusters. Usually, the generative process is $c_i \sim p(c_i|\alpha_1)$, $\boldsymbol{\mu}_k \sim p(\boldsymbol{\mu}_k|\alpha_2)$, $\mathbf{x}_i \sim p(\mathbf{x}_i|\boldsymbol{\mu}_{c_i})$, here $c_i \in \{1, 2, \cdots, K\}$ encodes the cluster assignment of the data point $\mathbf{x}_i$. $\alpha_1$ and $\alpha_2$ are hyperparameters. The integer random variable $K$ indicates the number of distinct values among the sampled $c_i$, and $\boldsymbol{\mu}_k$ denotes a parameter controlling the distribution of the $k$-th cluster. Specifically, these models include Gaussian mixture model [14, 38], Dirichlet process mixture model [8] and etc.

**Point-wise Clustering Posterior** Probabilistic model for clustering is usually presented by sequential sampling procedure to generate clusters. One of the prototypical tools for nonparametric clustering modeling is the Dirichlet Process. It allows for a discrete distribution of observations drawn from an arbitrary base measure over the domain in such a way as that the marginals match draws from, while simultaneously obtaining a countable set of distinct data points. A useful view of the Dirichlet process mixture model is the Chinese restaurant metaphor [12], which sequentially computes the conditional probability of assigning the current data point to one of already constructed clusters or a new one. The cluster assignment for each data point can be sampled by $c_i \sim p(c_i|c_{1:i-1}, \mathcal{D})$. Here $c_{1:i-1} = \{c_1, c_2, \cdots, c_{i-1}\}$ indicates the existing clusters formed by already assigned data points $\{\mathbf{x}_1, \mathbf{x}_2, \cdots, \mathbf{x}_{i-1}\}$. Based on existing clusters $c_{1:i-1}$ and observations $\mathcal{D}$, the cluster assignment for the $i$-th data point is sampled.

Given $N$ data points $\mathbf{X} = \{\mathbf{x}_i\}_{i=1}^N$, it is natural to draw independent samples to form clusters using decomposition

$$p(c_{1:N}|\mathcal{D}) = p(c_1|\mathcal{D})p(c_2|c_1, \mathcal{D}) \cdots p(c_N|c_{1:N-1}, \mathcal{D}) \tag{1}$$

After all data points are assigned, the final clusters are formed. Inference proceeds by traversing data points and re-sampling their cluster assignments in sequence.

**Challenges** The above process sequentially computes the conditional probability of assigning the current data point to one of already constructed clusters or a new one, and does not posit any particular prior on partitions. The sequential sampling procedure makes it process data points one by one, which limits its scalability on large datasets (the computational cost for full i.i.d. sampling $\{c_1, c_2, \cdots, c_N\}$ is high) [13]. Furthermore, the clustering result is sensitive to the sequential processing order. It needs a sufficient number of random samples to obtain stable result, which is time-consuming.

### 3.2 Amortized Mixing Coupling Processes

For the seek of effective cluster generative process and preventing the effect of sequential processing order, we focus on an alternative of clustering generative process from the view of cluster rather than the view of data point [27, 25]. Let $\mathcal{C} = \{\mathcal{C}_1, \mathcal{C}_2, \cdots, \mathcal{C}_K\}$ indicate $K$ clusters, we define $\mathcal{C}_k$

as $\mathcal{C}_k = (\mathcal{C}_k^{(1)}, \cdots, \mathcal{C}_k^{(i)}, \cdots, \mathcal{C}_k^{(N_k)})$, where $\mathcal{C}_k^{(i)}$ is the $i$-th data point of the $k$-th cluster. $N_k$ is the number of data points belonging to the $k$-th cluster. Based on the above definition, we are interested in sampling each cluster assignment by $\mathcal{C}_k \sim p(\mathcal{C}_k|\mathcal{C}_{1:k-1}, \mathcal{S}_k), k = 1, \cdots, K$, where $\mathcal{S}_k$ indicates the conditional information containing $M_k$ available unassigned data points for constructing the $k$-th cluster. $M_k$ is the number of left data points after generating the previous $k-1$ clusters, defined by $M_k = N - \sum_{j=1}^{k-1} N_j$ ($N_j$ is the number of data points belonging to the $j$-th cluster).

We formulate cluster assignment process in the form of *cluster-wise* ($\mathcal{C}_1 \to \mathcal{C}_2 \to \cdots \to \mathcal{C}_K$) rather than *point-wise* ($c_1 \to c_2 \to \cdots \to c_N$), which enables an efficient clustering process. We are interested in sampling $\{\mathcal{C}_1, \mathcal{C}_2, \cdots, \mathcal{C}_K\}$ via a decomposition as follows:

$$p(\mathcal{C}_{1:K}|\mathcal{D}) = p(\mathcal{C}_1|\mathcal{S}_1)p(\mathcal{C}_2|\mathcal{C}_1, \mathcal{S}_2) \cdots p(\mathcal{C}_K|\mathcal{C}_{1:K-1}, \mathcal{S}_K), \tag{2}$$

which is easy to model cluster sequentially, and the arbitrary number of clusters can be adaptively determined until there is no remaining point in $\mathcal{S}_k$. The sampling process defined in Eq.(2) indicates a clustering posterior using $K$ sets of indices.

To model the above cluster-wise sequential posterior, we propose a cluster-wise amortized mixing coupling processes (AMCP), which is able to provides an efficient amortized clustering architecture by grouping data points in a cluster-wise view rather than point-wise view. As shown in Figure 1(a), AMCP generates clusters sequentially and two key strategies, *Intra-Cluster Mixing* (*IntraCM*) and *Inter-Cluster Coupling* (*InterCC*), are proposed. Specifically, as shown in Figure 1, *IntraCM* focus on investigating the relationship between data points and reference distribution $\phi^{(k,T)} = [\phi_1^{(k,T)}, \phi_2^{(k,T)}, \cdots, \phi_k^{(k,T)}]$ with $\phi_j^{(k,T)} \in \mathbb{R}^d$ in a linear optimal transport view [33] with $T$ inner update steps. Then, *InterCC* leverages new insights on defining dissimilarity between unassigned data points in $\mathcal{S}_k$ and assigned data points in $\{\mathcal{C}_1, \mathcal{C}_2, \cdots, \mathcal{C}_{k-1}\}$ based on the learned optimal transport plan $\mathbf{Q}^{(k,T)}$ obtained in *IntraCM* in a coupling manner, which is able to select the dissimilar data points to form the next cluster $\mathcal{C}_k$ and update existing clusters $\{\mathcal{C}_1, \mathcal{C}_2, \cdots, \mathcal{C}_{k-1}\}$.

### 3.2.1 Intra-Cluster Mixing

We present intra-cluster mixing (*IntraCM*) from generative mixing view, a novel and fast framework for embedding each cluster in a vector space and defining the relations of each data point from the reference distribution, which give a clear evidence or constraint that the learned distributions could describe the cluster's summary statistics well, as shown in Figure 1(b). Specifically, in the $k$-th cluster generation step, data points are partitioned into several subsets, i.e., $\mathcal{D}^{(k)}$ has structures with $\{\mathcal{C}_1, \mathcal{C}_2, \cdots, \mathcal{C}_{k-1}, \mathcal{S}_k\}$. We model each data point $\mathbf{x}_i \in \mathcal{D}^{(k)}$ as a Bayesian mixture model with $k$ component parametrized by reference vectors $\phi^{(k)} = [\phi_1^{(k)}, \phi_2^{(k)}, \cdots, \phi_k^{(k)}] \in \mathbb{R}^{k \times d}$ and non-fixed mixing proportions over $k-1$ clusters $\{\mathcal{C}_1, \mathcal{C}_2, \cdots, \mathcal{C}_{k-1}\}$ and unassigned data points $\mathcal{S}_k$, as shown in Figure 1(a). A differentiable non-linear function $f_\theta(\cdot)$ (a neural network) is used to transform these representations $\phi_j^{(k)}$ into parameters $f_\theta(\phi_j^{(k)})$. By introducing latent variable $b_{i,j}^{(k)}$ indicating the data point $\mathbf{x}_i$ comes from the $j$-th component, we can get the full mixing likelihood related to the $k$-th cluster generation as follows:

$$p(\mathcal{D}^{(k)}|\phi^{(k)}) = \prod_{\mathcal{D}^{(k)}} \sum_{\mathbf{b}_i^{(k)}} p\left(\mathbf{x}_i, \mathbf{b}_i^{(k)}|f_\theta(\phi^{(k)})\right) = \prod_{\mathcal{D}^{(k)}} \sum_{j=1}^k \underbrace{p(b_{i,j}^{(k)})}_{\pi_j^{(k)}} p\left(\mathbf{x}_i|f_\theta(\phi_j^{(k)}), b_{i,j}^{(k)}\right), \tag{3}$$

where $\mathbf{b}_i^{(k)} = [b_{i,1}^{(k)}, b_{i,2}^{(k)}, \cdots, b_{i,k}^{(k)}]$ encodes the data point assignments. $\boldsymbol{\pi}^{(k)} = [\pi_1^{(k)}, \pi_2^{(k)}, \cdots, \pi_k^{(k)}]$ are the mixing coefficients.

However, directly optimizing $\log \prod_{\mathcal{D}^{(k)}} p(\mathbf{x}_i|f_\theta(\phi^{(k)}))$ with respect to $\phi^{(k)}$ is difficult due to marginalization over $\mathbf{b}_i^{(k)}$, while for many distributions optimizing $\log \prod_{\mathcal{D}^{(k)}} p(\mathbf{x}_i, \mathbf{b}_i^{(k)}|f_\theta(\phi_j^{(k)}))$ is much easier. In order to reflect the relationship of data in different sets, we introduce a variational distribution $q_{i,j}^{(k)} = q(b_{i,j}^{(k)}|\mathbf{x}_i)$ indicating a variational probability that the data point $\mathbf{x}_i$ belongs to the $j$-th component. In this case, we introduce a constrained variational lower bound to satisfy the optimal transport measure among different components and focus on performing the expected log likelihood by optimizing a lower bound:

$$\sum_{\mathcal{D}^{(k)}} \log \sum_{j=1}^k p\left(\mathbf{x}_i, b_{i,j}^{(k)}|f_\theta(\phi^{(k)})\right) \geq \sum_{\mathcal{D}^{(k)}} \sum_{j=1}^k q(b_{i,j}^{(k)}|\mathbf{x}_i) \log \frac{p\left(\mathbf{x}_i, b_{i,j}^{(k)}|f_\theta(\phi^{(k)})\right)}{q(b_{i,j}^{(k)}|\mathbf{x}_i)}$$

$$=: J(\phi^{(k)}, \mathbf{Q}^{(k)}) \tag{4}$$

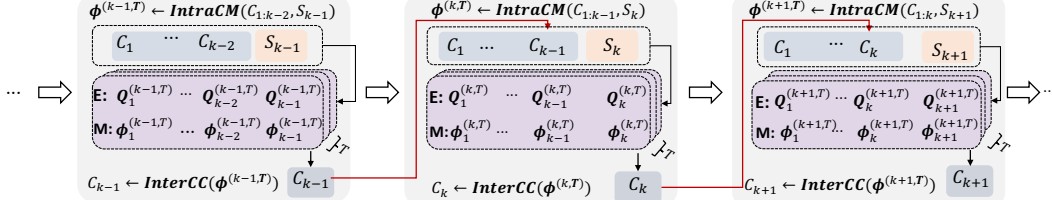

(a) The architecture of the proposed cluster-wise AMCP including two key strategies: Intra-cluster mixing (IntraCM) and inter-cluster coupling (InterCC). IntraCM and InterCC performs to generate cluster one by one.

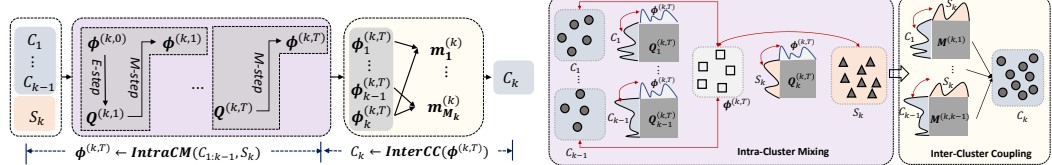

(b) Intra-cluster mixing and inter-cluster coupling. (c) Optimal transport view of intra-cluster mixing
Among them, IntraCM cooperates optimal transport and inter-cluster coupling, which focus on embedding from mixture view and describe the cluster's summary each cluster in a vector space and defining the rela- statistics with the aid of multiple EM steps. Then, In- tions of each data points from the reference distribu- terCC regarding the output of IntraCM as coupling to tion. generate the next cluster.

Figure 1: The architecture of the proposed AMCP.

**Optimal Transport View of *IntraCM*** Here we give a detailed analysis to connect the proposed mixing strategy with optimal transport measure [33, 11, 17], which defines the transport plan between data points as weighted points clouds or discrete measures. We aim to formulate an optimization problem for $q(b_{i,j}^{(k)}|\mathbf{x}_i)$ given in Eq.(4). By replacing the variational distribution to a joint distribution, i.e., $q_{i,j}^{(k)} = q(b_{i,j}^{(k)}|\mathbf{x}_i)q(\mathbf{x}_i)$, the optimization problem of maximizing the lower bound in Eq.(4) with respect to the transport plan matrix $\mathbf{Q}^{(k)} = (q_{i,j}^{(k)}) \in \mathbb{R}^{N \times k}$ is equivalent to solving following entropic regularized Kantorovich relaxation of optimal transport [28]:

$$J(\boldsymbol{\phi}^{(k)}, \mathbf{Q}^{(k)}) = \min_{\mathbf{Q}^{(k)}} \sum_{\mathcal{D}^{(k)}} \sum_{j=1}^{k} -\log p\left(\mathbf{x}_i, b_{i,j}^{(k)}|f_\theta(\boldsymbol{\phi}^{(k)})\right) q_{i,j}^{(k)} - \epsilon H(\mathbf{Q}^{(k)}), \tag{5}$$

where $-\log p(\mathbf{x}_i, b_{i,j}^{(k)}|f_\theta(\boldsymbol{\phi}^{(k)}))$ indicates the pairwise transport cost between data point and reference vector. If *Gaussian* distribution is used as data likelihood, the transport cost $-\log p(\mathbf{x}_i, b_{i,j}^{(k)}|f_\theta(\boldsymbol{\phi}^{(k)}))$ is equivalent to a negative kernel $-\mathcal{K}(\mathbf{x}_i, f_\theta(\boldsymbol{\phi}_j^{(k)}))$. The $\epsilon$ impacted entropic regularized term $H(\mathbf{Q}^{(k)}) = -\sum_{i,j} q_{i,j}^{(k)} \log q_{i,j}^{(k)}$ controls the sparsity of $\mathbf{Q}^{(k)}$.

In original entropic regularized Kantorovich relaxation of optimal transport [28], there are some constraints on transport plan $\mathbf{Q}^{(k)}$, i.e., $\sum_{i=1}^{N} q_{i,j}^{(k)} = 1/k$ and $\sum_{j=1}^{k} q_{i,j}^{(k)} = 1/N$. If we set the factorized variational distribution to $q(\mathbf{x}_i) = 1/N$, the constraint $\sum_j q_{i,j}^{(k)} = 1/N$ is automatically met. Optimal transport view induces various advantages of our proposed mixing strategy. First, it is is computationally efficient, scalable to vary large dataset. Unlike the quadratic cost of traditional similarity weight computation (e.g., the attention score in self-attention), our optimal transport mixing uses a fixed number ($k$) of references serving as queries and the number of references is set depends on data without manual setting. Secondly, it allows unsupervised learning where the model parameters $\{\boldsymbol{\phi}_j^{(k)}\}_{j=1}^k$ can be learned without target (class) labels. Third, it is able to sufficiently exploit pairwise interactions between data points in intra-cluster level and benefit for the next inter-cluster coupling, which is useful to sufficiently mine the hidden structure among data.

**Expectation Maximization for *IntraCM*** Considering the latent variable $\mathbf{b}^{(k)}$ is introduced, naturally the expectation maximization (EM) method can be utilized [11]. Iterative optimization of this bound alternates between two steps:

- **E-step**: The E-step focuses on maximizing the above bound with respect to the variational probability $q(b_{i,j}^{(k,t)}|\mathbf{x}_i)$. We compute a new estimate of the posterior probability distribution over the

latent variables from the previous iteration, which yields a soft-assignment of the data points to the components $q(b_{i,j}^{(k,t)}|\mathbf{x}_i) = \frac{\pi_j^{(k,t-1)} p\left(\mathbf{x}_i|b_{i,j}^{(k,t-1)}, f_\theta(\phi^{(k,t-1)})\right)}{\sum_{j=1}^{k} \pi_j^{(k,t-1)} p\left(\mathbf{x}_i|b_{i,j}^{(k,t-1)}, f_\theta(\phi^{(k,t-1)})\right)}$, where $\pi_j^{(k,t-1)}$ and $\phi^{(k,t-1)}$ indicates model parameters from previous iteration. This soft-assignment is a regular E-step without the balanced assignment constraints. Furthermore, the variational lower bound can be formulated as the optimal transport setting (as shown in Eq.(5)), yielding an Sinkhorn-based optimal transport solution [30] as follows:

$$q(b_{i,j}^{(k,t)}|\mathbf{x}_i) = u_i \exp\left(p\left(\mathbf{x}_i|b_{i,j}^{(k,t-1)}, f_\theta(\phi^{(k,t-1)})\right)/\epsilon\right) v_j, \tag{6}$$

where $u_i = \frac{1}{N\sum_{j=1}^{k} \exp(p(\mathbf{x}_i|b_{i,j}^{(k,t-1)}, f_\theta(\phi^{(k,t-1)}))/\epsilon)v_j}$ and $v_j = \frac{1}{k\sum_{i=1}^{N} \exp(p(\mathbf{x}_i|b_{i,j}^{(k,t-1)}, f_\theta(\phi^{(k,t-1)}))/\epsilon)u_i}$. Note that above solution usually converges quickly after a few iterations. Hence, the E-step coincides with the optimal transport based similarity weight computation, which leads to the various advantages of our proposed mixing strategy and forms the transport plan $\mathbf{Q}^{(k,t)}$ for all data points.

- **M-step**: Once the E-step is done (i.e., $\mathbf{Q}^{(k,t)}$ is found), we perform the M-step, that is, maximizing the lower bound in Eq.(4) with respect to $\phi^{(k,t)}$ when $\mathbf{Q}^{(k,t)}$ is fixed. Due to the non-linearity of $f_\theta(\cdot)$ there exists no analytical solution to $\arg\max_{\phi^{(k)}} J(\phi^{(k)}, \mathbf{Q}^{(k,t)})$. With the benefit of differentiable $f_\theta(\cdot)$, we can solve it with the aid of gradient-based strategy. Assume that likelihood $p(\mathbf{x}_i|f_\theta(\phi_j^{(k)}), b_{i,j}^{(k)})$ follows a *Gaussian* distribution with fixed covariance, we can obtain following update rule:

$$\phi_j^{(k,t)} = \phi_j^{(k,t-1)} + \eta \sum_{\mathcal{D}^{(k)}} q(b_{i,j}^{(k,t)}|\mathbf{x}_i)\left(f_\theta(\phi_j^{(k,t-1)}) - \mathbf{x}_i\right)\nabla f_\theta(\phi_j^{(k,t-1)}). \tag{7}$$

For simplicity, if we assume the parameters $\phi^{(k)}$ in likelihood with a linear transformation $f_\theta(\phi^{(k)}) = \mathbf{W}\phi^{(k)}$, it can yield an optimal transport kernel embedding solution [26] when we set it as *Gaussian* distribution with fixed variance, e.g., $\pi_j^{(k,t)} = \frac{\sum_i q(b_{i,j}^{(k,t-1)}|\mathbf{x}_i)}{N}$ and $\phi_j^{(k,t)} = \sum_i \frac{\exp(\phi_j^{(k,t-1)\top}(\mathbf{W}\mathbf{x}_i)/\sqrt{d})}{\sum_i \exp(\phi_j^{(k,t-1)\top}(\mathbf{W}\mathbf{x}_i)/\sqrt{d})}\mathbf{W}\mathbf{x}_i$, which is equivalent to induced self-attention introduced in Lee et al. [20] providing an efficient solution for pairwise weights measuring.

Usually, we can take multiple EM steps for further optimization (e.g., $\phi^{(k,0)} \to \phi^{(k,1)} \to \cdots \to \phi^{(k,T)}$). Here $T$ is the number of EM steps. *IntraCM* belongs to the class of generalized EM algorithms and is guaranteed to converge to a local optimum of the data log likelihood [35].

### 3.2.2 Inter-Cluster Coupling

After $T$ EM steps, the variational probability $q_{i,j}^{(k,T)} = q(b_{i,j}^{(k,T)}|\mathbf{x}_i)$ can be regarded as the optimal transport plan, which carries the information on how to distribute the mass of data points $\mathbf{x}_i$ to reference vectors $\phi_j^{(k,T)}$. By regarding $\phi_j^{(k,T)}$ as couplings between unassigned set and assigned set, we propose inter-cluster coupling (*InterCC*) to generate the next cluster, as shown in Figure 1. Considering the current data points have structures with $k$ different sets, $\mathcal{C}_1, \mathcal{C}_2, \cdots, \mathcal{C}_{k-1}, \mathcal{S}_k$, we can get $k-1$ transport plan matrices $\{\mathbf{U}_1^{(k)}, \mathbf{U}_2^{(k)}, \cdots, \mathbf{U}_{k-1}^{(k)}\}$ related to $k-1$ existing clusters and $\mathbf{V}^{(k)}$ related to the unassigned set $\mathcal{S}_k$,

$$\mathbf{U}_l^{(k)} = [q(b_{i,j}^{(k,T)}|\mathbf{x}_i \in \mathcal{C}_l)]^{|\mathcal{C}_l|\times k} \quad l = \{1, 2, \cdots k\} \quad \text{and} \quad \mathbf{V}^{(k)} = [q(b_{i,j}^{(k,T)}|\mathbf{x}_i \in \mathcal{S}_k)]^{M_k \times k}. \tag{8}$$

We can measure the relations between unsigned data points in $\mathcal{S}_k$ and existing assigned data points in $k-1$ existing clusters $\mathcal{C}_1, \mathcal{C}_2, \cdots, \mathcal{C}_{k-1}$ with the aid of couplings $\phi^{(k)}$ as follows

$$\mathbf{M}^{(k)} = \Big[\underbrace{\oplus_{l=1}^{k-1}(\mathbf{U}_l^{(k)}\phi^{(k,T)}\mathbf{v}_1^{(k)\top})}_{\mathbf{m}_1^{(k)}}, \cdots, \underbrace{\oplus_{l=1}^{k-1}(\mathbf{U}_l^{(k)}\phi^{(k,T)}\mathbf{v}_{M_k}^{(k)\top})}_{\mathbf{m}_{M_k}^{(k)}}\Big] \in \mathbb{R}^{(N-M_k)\times M_k}, \tag{9}$$

where $\oplus$ is a column-wise concatenation operation and $\mathbf{v}_i^{(k)}$ is the $i$-th column of $\mathbf{V}^{(k)}$. It is reasonable to select dissimilar data point in unassigned data points set $\mathcal{S}_k$ to form a new cluster $\mathcal{C}_k$, i.e.,

$$\mathcal{C}_k \sim \text{MULT}\left(\text{SOFTMAX}\left(\Big[1 - \mathbf{I}^\top \mathbf{m}_1^{(k)}, 1 - \mathbf{I}^\top \mathbf{m}_2^{(k)}, \cdots, 1 - \mathbf{I}^\top \mathbf{m}_{M_k}^{(k)}\Big]\right)\right). \tag{10}$$

Here *Multinomial* distribution is used to model the generation of $\mathcal{C}_k$. $\mathbf{I} \in \mathbb{R}^{N-M_k}$ is a vector of whose elements are all 1s. $\mathbf{m}_j^{(k)}$ is the $i$-th column of $\mathbf{M}^{(k)}$. Thus, the data points related to dissimilar

indices in each matrix are selected to generate the new cluster $\mathcal{C}_k$. Furthermore, previous existing clusters $\mathcal{C}_1, \mathcal{C}_2, \cdots, \mathcal{C}_{k-1}$ can be updated based on dissimilarity matrix. For a given cluster $\mathcal{C}_l$, it can be updated by $\mathcal{C}_l := \mathcal{C}_l \cup \mathcal{C}_l^{\text{new}}$, where $\mathcal{C}_l^{\text{new}} := \text{MULT}(\text{SOFTMAX}[1 - \mathbf{I}^{(l)^\top} \mathbf{M}^{(k,l)}])$. $\mathbf{I}^{(l)} \in \mathbb{R}^{|\mathcal{C}_l|}$ is a vector of all 1s and $\mathbf{M}^{(k,l)} = \mathbf{U}_l^{(k)} \boldsymbol{\phi}^{(k,T)} \mathbf{V}^{(k)^\top}$.

### 3.2.3  Iterative Unrolling Inference

*IntraCM* and *InterCC* are both differentiable procedures for optimal transport measuring and next cluster constructing, whose outcome relies on the current existing clusters and the statistical model parameters $\boldsymbol{\phi}^{(k)}$. As shown in Eq.(2), the whole cluster-wise clustering procedure can be regarded as a recurrent *IntraCM* with *InterCC* until there are no available data point left. In this case, the information about statistical regularities in our model that is required for generating the sequential clusters is encoded in a series of parameters $\{\boldsymbol{\phi}^{(1,T)}, \boldsymbol{\phi}^{(2,T)}, \cdots, \boldsymbol{\phi}^{(K,T)}\}$. By unrolling the iterations of the presented mixing process, we obtain an end-to-end differentiable clustering procedure based on the statistical model parameterized by $\{\boldsymbol{\phi}^{(1,T)}, \boldsymbol{\phi}^{(2,T)}, \cdots, \boldsymbol{\phi}^{(K,T)}\}$. We can therefore use (stochastic) gradient descent and fit the statistical model to capture the regularities corresponding to clusters for a given dataset.

$$\mathbb{E}_{p(\mathcal{C}_{1:K}, \mathcal{D})} \log p_\theta(\mathcal{C}_{1:K}|\mathcal{D}) \geq \mathbb{E}_{p(\mathcal{C}_{1:K}, \mathcal{D})} \sum_{k=1}^{K} \left[ \underbrace{\sum_{j=1}^{k} q(b_{i,j}^{(k)}|\mathbf{x}_i) \log \frac{p\left(\mathbf{x}_i, b_{i,j}^{(k)}|f_\theta(\boldsymbol{\phi}^{(k)})\right)}{q(b_{i,j}^{(k)}|\mathbf{x}_i)}}_{\textit{IntraCM}} + \underbrace{p(\mathcal{C}_k|f_\theta(\boldsymbol{\phi}^{(k)}))}_{\textit{InterCC}} \right]$$

$$(11)$$

The likelihood is derived from approximating the $p_\theta(\mathcal{C}_{1:K}|\mathcal{D})$ to true posterior via KL divergence. Note that samples from $p(\mathcal{C}_{1:K}, \mathcal{D})$ are obtained from the explicit generative model, such as *Gaussian* mixture model, *Dirichlet* process mixture model. With infinite generated samples, we can train a model to approximate $p_\theta(\mathcal{C}_{1:K}|\mathcal{D})$ accurately. The whole inference procedure is given in Algorithm 1. By repeating the cluster generative process until there are no data points left, IntraCM and InterCC are processed alternatively, and the clusters are iteratively refined by deriving the explicit supervisory signal from the already formed clusters. Note that the complexity of traditional methods measuring weights between data points acquires $O(KN^2)$ (e.g., self-attention) which may be too expensive for large sets, and our AMCP only acquires $O(KNk)$ in the EM-step for the $k$-th iteration, which is more efficient.

---

**Algorithm 1** Iterative Unrolling Inference for for AMCP

---

**Input:** $\mathcal{D} = \{\mathbf{x}_1, \mathbf{x}_2, \cdots, \mathbf{x}_N\}$ with $N$ data points. Randomly initialized parameters $\{\boldsymbol{\phi}^{(k,0)}\}_{k=1}^{K}$.
1: Establish the first cluster $\mathcal{C}_1$ by selecting cluster with the highest probability from $k$-means or GMM
2: **while** not new cluster $\mathcal{C}_k$ is generated **do**
3:     // Intra-Cluster Mixing
4:     Establish expected lower bound $J(\boldsymbol{\phi}^{(k,0)}, \mathbf{Q}^{(k,0)})$ via Eq.(4)
5:     **for** $t = 1, 2, \cdots, T$ **do**                                          ▷ Multiple EM steps
6:         $\mathbf{Q}^{(k,t)} \leftarrow J(\boldsymbol{\phi}^{(k,t-1)}, \mathbf{Q}^{(k,t-1)})$, $\quad \boldsymbol{\phi}^{(k,t)} \leftarrow \arg\max_{\boldsymbol{\phi}^{(k)}} J(\boldsymbol{\phi}^{(k,t-1)}, \mathbf{Q}^{(k,t)})$        ▷ EM-step
7:     **end for**
8:     // Inter-Cluster Coupling
9:     Establish $k$ transport plan matrices $\{\mathbf{U}_l^{(k)}\}_{l=1}^{k-1}$ and $\mathbf{V}^{(k)}$, and similarity matrix $\mathbf{M}^{(k)}$
10:     Generate $\mathcal{C}_k$
11:     Update existing clusters $\mathcal{C}_1, \mathcal{C}_2, \cdots, \mathcal{C}_{k-1}$                                   ▷ Optional
12:     **if** $\mathcal{S}_k \neq \emptyset$ **then**
13:         Establish unassigned set $\mathcal{S}_{k+1}$
14:     **end if**
15: **end while**
**Output:** updated parameters $\boldsymbol{\phi}^{(K,T)}$, generated clusters $\mathcal{C}_1, \mathcal{C}_2, \cdots, \mathcal{C}_K$

---

## 4  Experiments

In this section, we evaluate the proposed AMCP in terms of synthetic and real-world datasets by comparing with the state-of-the-art methods.

### 4.1  Experiments on Synthetic Datasets

The first experiment was conducted on synthetic data with arbitrary number of clusters. Thus, five existing methods handling variable number of clusters, MCMC[1], VI[2], DAC[21], NCP[27] and

Table 1: Clustering performance on synthetic 2D GMM. The numbers below Scenario 1 and 2 are oracle *LL* values computed by the true parameters. The average on 5 runs are reported.

| Scenario | Metric | MCMC [1] | VI [2] | DAC [21] | NCP [27] | ST-ACT [20] | AMCP |
|---|---|---|---|---|---|---|---|
| *Scenario 1* (-1.2452) | *LL* | $-1.5111_{\pm0.021}$ | $-1.4230_{\pm0.081}$ | $\underline{-1.2671_{\pm0.041}}$ | $-1.3251_{\pm0.042}$ | $-1.2928_{\pm0.051}$ | $\mathbf{-1.2517_{\pm0.035}}$ |
| | *ACC* | $0.7066_{\pm0.003}$ | $0.6832_{\pm0.004}$ | $0.7923_{\pm0.001}$ | $0.8131_{\pm0.002}$ | $0.7680_{\pm0.003}$ | $\mathbf{0.8532_{\pm0.002}}$ |
| | *NMI* | $0.8731_{\pm0.004}$ | $0.8531_{\pm0.002}$ | $0.8983_{\pm0.003}$ | $\underline{0.8923_{\pm0.004}}$ | $0.9123_{\pm0.003}$ | $\mathbf{0.9334_{\pm0.001}}$ |
| | Time[s] | $0.0832_{\pm0.032}$ | $0.0332_{\pm0.045}$ | $0.0124_{\pm0.047}$ | $0.0343_{\pm0.026}$ | $\underline{0.0114_{\pm0.002}}$ | $\mathbf{0.0113_{\pm0.003}}$ |
| *Scenario 2* (-1.8242) | *LL* | $-1.924_{\pm0.022}$ | $-2.9026_{\pm0.022}$ | $\underline{-1.847_{\pm0.034}}$ | $-1.8551_{\pm0.042}$ | $-1.8918_{\pm0.051}$ | $\mathbf{-1.8420_{\pm0.025}}$ |
| | *ACC* | $0.6531_{\pm0.002}$ | $0.5944_{\pm0.003}$ | $\underline{0.7021_{\pm0.002}}$ | $0.7233_{\pm0.002}$ | $0.6926_{\pm0.002}$ | $\mathbf{0.7574_{\pm0.002}}$ |
| | *NMI* | $0.7931_{\pm0.005}$ | $0.8123_{\pm0.003}$ | $0.8231_{\pm0.002}$ | $\underline{0.8089_{\pm0.004}}$ | $0.8412_{\pm0.002}$ | $\mathbf{0.8592_{\pm0.002}}$ |
| | Time[s] | $0.0953_{\pm0.053}$ | $0.0362_{\pm0.056}$ | $\underline{0.0151_{\pm0.044}}$ | $0.0366_{\pm0.035}$ | $0.0166_{\pm0.001}$ | $\mathbf{0.0124_{\pm0.002}}$ |

ST-ACT[20] are used as baselines. Among them, MCMC and VI are traditional approximation methods, and DAC, NCP and ST-ACT are deep amortized clustering methods. The synthetic data is generated via a 2D *Gaussian* mixture model (GMM) which is defined by the following process $\alpha \sim \text{EXP}(1)$, $c_{1:N} \sim \text{CRP}(\alpha)$, $\boldsymbol{\mu}_k \sim \mathcal{N}(0, \sigma_\mu^2 \mathbf{1})$, $\mathbf{x}_i \sim \mathcal{N}(\boldsymbol{\mu}_{c_i}, \sigma^2 \mathbf{I})$. We set $\sigma_\mu = \sigma = 10$. At each training step, we generate 10 random datasets according to the above generative process. Each dataset contains 200 points on a 2D plane, and each sampled from one of 4 Gaussians. Two testing scenarios (*Scenario 1* and *Scenario 2*) are constructed to evaluate the effectiveness of the proposed method. The test set in *Scenario 1* has the same configuration (200 samples and 4 clusters) as training set, while *Scenario 2* contains different numbers of samples and clusters (400 samples and 6 clusters) in order to verify whether the clustering method can generalize to the unseen clusters. Table 1 summarizes the results in terms of two testing scenarios. Here we report log-likelihood (*LL*), clustering accuracy (*ACC*), Normalized Mutual Information (*NMI*) and running time (seconds). The best and second results are marked in bold and underlined. As expected, the proposed AMCP consistently outperforms baselines on both scenarios. Although *Scenario 2* is more challenging, amortized clustering methods (DAC, NCP, ST-ACT and our AMCP) can capture cluster uncertainty and obtain competitive results. For testing time, amortized methods are obviously faster than traditional sampling based method (MCMC). Although VI is much fast than MCMC, it is still lower than amortized methods. In cluster-wise amortized clustering methods, DAC, ST-ACT and our ACMP achieve competitive running time since cluster-wise sequential modeling is more efficient than point-wise sequential modeling.

### 4.2 Experiments on Real-world Datasets

Real-world datasets, *MNIST* [19], *Tiny-ImageNet* [18] and *CIFAR-10* [34], are used to validate the performance. To formulate the amortized mechanism, we generate training and test data via following *Dirichlet* process mixture model: $\alpha \sim \text{EXP}(1)$, $c_{1:N} \sim \text{CRP}(\alpha)$, $K - 1 \sim \text{BINOMIAL}(K_g - 1, 0.5)$, $l_k \sim U(0, K-1)$, $\mathbf{x}_i \sim U[\mathcal{D}, l_{c_i}]$. Here $\mathbf{x}_i \sim U[\mathcal{D}, l_{c_i}]$ indicates sampling $\mathbf{x}_i$ with label $l_{c_i}$ uniformly from dataset $\mathcal{D}$, which can be training set or test set. $K_g$ indicates the number of clusters we can sample. For all original dataset, data containing half of the classes is used to generate the training sets, and the remaining half is used to generate the test sets, with no overlap between training classes and test classes. We sample multiple clustered datasets to form training sets with $N \in \{1000, 10000\}$. For test set, we generate 1000 randomly clustered datasets with $N \in \{1000, 5000\}$. More information is given in Appendix.

In Table 2, we depict the quantitative clustering results in real-world datasets. The best and second results are marked in bold and underlined. The average clustering results are recorded over 5 runs with different random parameter initialization. It can be seen that our AMCP outperforms all baselines. The main reason, we believe, is that AMCP not only exploits amortized properties between data points in both inter- and intra-clusters, but also has flexible clustering process, which simultaneously generate the new cluster and update the summary statistics for previous clusters. In model efficiency evaluation, the DAC, ST-ACT and AMCP consume a more competitive running time, which outperform other methods by a large margin. The reason is that DAC and ST-ACT identify each cluster after one forward pass, and AMCP generates clusters in cluster-wise manner and exploits data relations via the efficient optimal transport. However, DAC, ST-ACT are pretty worse than AMCP on clustering performance.

Among amortized clustering methods, DAC, NCP, ST-ACT and AMCP are able to process arbitrary number of clusters. For better visualization of cluster generative process, a digit subset from *MNIST* test set is randomly sampled. To illustrate how the clustering models capture the shape ambiguity

Table 2: Clustering performance on real-world datasets. The average on 5 runs are reported.

| Dataset | Metric | MCMC [1] | VI [2] | DAC [21] | NCP [27] | ST-ACT [20] | AMCP |
|---|---|---|---|---|---|---|---|
| *MNIST* | ACC | $0.8630_{\pm0.004}$ | $0.9758_{\pm0.004}$ | $0.9796_{\pm0.001}$ | $0.9633_{\pm0.004}$ | $0.9596_{\pm0.003}$ | $\mathbf{0.9845}_{\pm0.001}$ |
| | NMI | $0.9124_{\pm0.005}$ | $0.9482_{\pm0.002}$ | $0.9542_{\pm0.001}$ | $0.9321_{\pm0.002}$ | $0.9465_{\pm0.002}$ | $\mathbf{0.9624}_{\pm0.001}$ |
| | Time[s] | $56.33_{\pm6.42}$ | $63.46_{\pm6.42}$ | $24.33_{\pm3.53}$ | $73.34_{\pm2.42}$ | $22.56_{\pm2.57}$ | $\mathbf{18.22}_{\pm3.74}$ |
| *Tiny-ImageNet* | ACC | $0.0637_{\pm0.005}$ | $0.0785_{\pm0.001}$ | $0.0907_{\pm0.004}$ | $0.1028_{\pm0.002}$ | $0.0892_{\pm0.003}$ | $\mathbf{0.1324}_{\pm0.002}$ |
| | NMI | $0.3212_{\pm0.002}$ | $0.3231_{\pm0.004}$ | $0.2894_{\pm0.005}$ | $0.3254_{\pm0.002}$ | $0.2832_{\pm0.003}$ | $\mathbf{0.3421}_{\pm0.003}$ |
| | Time[s] | $3140.8_{\pm18.26}$ | $3450.5_{\pm19.15}$ | $1415.6_{\pm16.57}$ | $4732.8_{\pm32.11}$ | $1316.6_{\pm21.15}$ | $\mathbf{1186.4}_{\pm19.37}$ |
| *CIFAR-10* | ACC | $0.6322_{\pm0.005}$ | $0.8742_{\pm0.002}$ | $0.8835_{\pm0.003}$ | $0.7641_{\pm0.005}$ | $0.8632_{\pm0.003}$ | $\mathbf{0.9024}_{\pm0.002}$ |
| | NMI | $0.6782_{\pm0.004}$ | $0.7765_{\pm0.004}$ | $0.7873_{\pm0.002}$ | $0.7431_{\pm0.004}$ | $0.7673_{\pm0.005}$ | $\mathbf{0.8042}_{\pm0.001}$ |
| | Time[s] | $174.42_{\pm6.12}$ | $192.42_{\pm5.11}$ | $78.33_{\pm4.51}$ | $239.33_{\pm9.17}$ | $68.32_{\pm6.51}$ | $\mathbf{59.33}_{\pm5.59}$ |

Table 3: A toy clustering results comparisons among different amortized clustering methods (DAC, NCP, CHiGac and AMCP) on *MNIST* dataset.

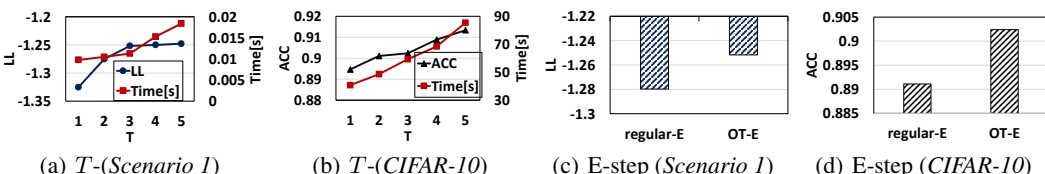

of some of the digits, we plot the ground-truth digit clusters and clustering results in Table 3. Comparing with DAC, NCP and ST-ACT, the proposed AMCP can get the most accurate results. For example, DAC assigns the digit 7 (with similar appearance to 9) to cluster 9, and NCP generates a new cluster for it. Fortunately, AMCP correctly assigns it to cluster 7. All baselines are unable to assign the digit 4 written in a strange way to the right cluster, and our AMCP correctly assigns it to cluster 4. Results on *Tiny-ImageNet* and *CIFAR-10* can be obtained in Appendix.

### 4.3 Model Analysis

We empirically analyze the proposed model from the impact of various hyper-parameters. They are summarized as follows: (1)**The Number of EM Steps** $T$**:** AMCP can apply multiple EM steps to achieve more accurate $\phi^{(K,T)}$. Here we conduct experiments on varying $T$ in $\{1, 2, 3, 4, 5\}$ to investigate the effect of $T$. Figure 2(a) and (b) show the effect of $T$ on synthetic data (*Scenario 1*) and real-world data (*CIFAR-10*). We can see that, the performance (*LL* and *ACC*) become better and better with the increasing of $T$, while the running time is linearly scalable to the number of steps. (2) **OT E-step vs. Regular E-step** We introduce two different ways to conduct E-step in IntraCM: regular soft-assignment E-step and Sinkhorn-based optimal transport E-step. Figure 2(c) and (d) show the comparison between regular E-step and OT E-step on synthetic data (*Scenario 1*) and real-world data (*CIFAR-10*). The maximum number of Sinkhorn-Knopp iterations for solving the OT problem is set to 10. While the regular E-step is consistently faster than the OT E-step since it's closed-form solution, OT E-step can achieve more accurate likelihood and accuracy. Furthermore, we compare the difference between the strategy of updating the existing clusters and the strategy of not updating. More empirical results are given in Appendix.

(a) $T$-(*Scenario 1*)  (b) $T$-(*CIFAR-10*)  (c) E-step (*Scenario 1*)  (d) E-step (*CIFAR-10*)

Figure 2: The impact of hyper-parameters.

## 5 Conclusion

In this paper, we studied the problem of amortized clustering, and presented our approach that perform cluster-wise amortized mixing coupling processes via intra-cluster mixing and inter-cluster coupling. An interesting direction for future research is to explore more complicated and realistic

hidden structure (e.g., hierarchical clusters) that is not possible with our approach. Furthermore, it is interested to explore novel applications that can be enabled by the interpretability and controllability brought by the amortized clustering.

## 6 Acknowledgement

This work was partly supported by the Beijing Natural Science Foundation under Grant (Z180006, L211016); the National Key Research and Development Program (2020AAA0106800); the National Natural Science Foundation of China under Grant 62176020; CAAI-Huawei MindSpore Open Fund; and Chinese Academy of Sciences(OEIP-O-202004).

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
