# Appendix: Amortized Mixing Coupling Processes for Clustering

**Huafeng Liu**[1,2]**, Liping Jing**[1]

[1]Beijing Key Lab of Traffic Data Analysis and Mining, Beijing Jiaotong University, Beijing, China
[2]The Department of Mathematics, The University of HongKong, Hong Kong SAR, China
huafeng.liu@outlook.com, lpjing@bjtu.edu.cn

## A  Some Basic Concepts

Here, we revisit some concepts, which would help us understand our amortized mixing clustering process.

### A.1  Optimal Transport

Our intra-cluster mixing is based on the transport plan between input data point $\mathbf{x}_i$ and reference vectors $\boldsymbol{\phi} = [\boldsymbol{\phi}_1, \boldsymbol{\phi}_2, \cdots, \boldsymbol{\phi}_k]$ with $\boldsymbol{\phi}_j \in \mathbb{R}^d$ seen as discrete measures, which is a by-product of the optimal transport problem. Here we adopt the Kantorovich relaxation of optimal transport with entropic regularization [11]. Let $\mathbf{C} \in \mathbb{R}^{N \times k}$ be a matrix representing the pairwise costs for aligning the elements of $\mathbf{x}_i$ and $\boldsymbol{\phi}_j$. The entropic regularized Kantorovich relaxation of optimal transport from $\mathbf{x}_i$ to $\boldsymbol{\phi}_j$ focuses on learning transport plan weight matrix $\mathbf{Q}$ in following manner:

$$J(\mathbf{Q}) = \min_{\mathbf{Q}} \sum_{\mathcal{D}} \sum_{j=1}^{k} c_{i,j} q_{i,j} - \epsilon H(\mathbf{Q}), \quad \text{s.t.} \quad \sum_{i=1}^{N} q_{i,j} = \frac{1}{k}, \ \forall j \ \text{ and } \ \sum_{j=1}^{k} q_{i,j} = \frac{1}{N}, \ \forall i \quad (1)$$

where the entropic regularized term $H(\mathbf{Q}) = -\sum_{i,j} q_{i,j} \log q_{i,j}$ with parameter $\epsilon$ controls the sparsity of $\mathbf{Q}$. Typically, the cost matrix $\mathbf{C}$ is defined as the negative kernel $c_{i,j} = -\mathcal{K}(\mathbf{x}_i, \boldsymbol{\phi}_j)$. Usually, above problem can be solved a matrix scaling procedure: Sinkhorn's algorithm [12].

### A.2  Sinkhorn's Algorithm for Optimal Transport

Considering above entropic regularized Kantorovich relaxation of optimal transport (Eq. (1)), we can solve it efficiently by Sinkhorn's algorithm with a few fixed point matrix scaling iterations. Specifically, SK algorithm finds the optimal solution as

$$\mathbf{Q} = \text{diag}(\mathbf{u}^{(l)}) \mathbf{A} \text{diag}(\mathbf{v}^{(l)}) \quad (2)$$

where the elements in $\mathbf{A} \in \mathbb{R}^{N \times k}$ are defined as $a_{i,j} = \exp(-c_{i,j}/\epsilon)$ and

$$\mathbf{u}^{(l+1)} = \frac{1/N}{\mathbf{A}\mathbf{v}^{(l)}}, \quad \mathbf{v}^{(l+1)} = \frac{1/k}{\mathbf{A}\mathbf{u}^{(l)}} \quad (3)$$

Based on iterated updating, the matrix $\text{diag}(\mathbf{u}^{(l)}) \mathbf{A} \text{diag}(\mathbf{v}^{(l)})$ converges to $\mathbf{Q}$ after a few iterations. Since the iterated SK algorithm constitutes the feed-forward pass for the whole inference procedure, IntraCC can be seamlessly composed to the whole inference procedure to achieve efficient computation. Furthermore, all the operations above at each step are differentiable, which enables $\boldsymbol{\phi}$ to be optimized through back-propagation. Consequently, this module can be easily injected into the whole AMCP.

36th Conference on Neural Information Processing Systems (NeurIPS 2022).

## A.3 Inductive Biases

First, we give a concept of Permutation Invariant Function which is the basic property of stochastic process, e.g., neural processes and gaussian process.

**Definition 1.** *(Permutation Invariant Function) A function $f(\cdot) : \times_i^N \mathbb{R}^M \to \mathbb{R}^D$ mapping a set of data points $\{x_i\}_{i=1}^N$ is Permutation Invariant Function if*

$$\mathbf{x} = [x_1, x_2, \cdots, x_N] \to f = [f_1(x_{\pi(1:N)}), f_2(x_{\pi(1:N)}), f_D(x_{\pi(1:N)})] \tag{4}$$

*where $x_i \in \mathbb{R}^M$ and the function output is a D dimensional vector. Operation $\pi : [1, 2, \cdots, N] \to [\pi_1, \pi_2, \cdots, \pi_N]$ is a permutation set over the order of elements in the set.*

**Definition 2.** *(Permutation Equivariant Function) A function $f(\cdot) : \times_i^N \mathbb{R}^M \to \mathbb{R}^N$ mapping a set of data points $\{x_i\}_{i=1}^N$ is Permutation Invariant Function if*

$$\mathbf{x}_\pi = [x_{\pi_1}, x_{\pi_2}, \cdots, x_{\pi_N}] \to f_\pi = \pi \circ f(x_{1:N}) \tag{5}$$

*where the function output contains $N$ elements keeping the order of inputs.*

Permutation Equivariant Function keeps the order of elements in the output consistent with that in the input under any permutation operation $\pi$. Permutation invariant functions are candidate functions for learning embeddings of a set or other order uncorrelated data structure $\{x_i\}_{i=1}^N$, and the invariant property is easy to be verified. Here, we give a mean operation structure over the output

$$F(\mathbf{x}_{\pi(1:N)}) = \left( \frac{1}{N} \sum_{i=1}^N \phi_1(x_i), \frac{1}{N} \sum_{i=1}^N \phi_2(x_i), \cdots, \frac{1}{N} \sum_{i=1}^N \phi_M(x_i) \right) \tag{6}$$

## A.4 ST-ACT

The ST-ACT [7] is a set transformer based amortized clustering with Adaptive Computation Time (ACT) [4], which is able to identify the clusters iteratively and make it to learn when to stop. In Lee et al. [8]'s appendix, the authors give the detailed description of ST-ACT. By defining an adaptive-computation-time version of Point-inducing Multi-head Attention as follows:

$$aPMA(\mathbf{X}, k) = MAB([\mathbf{s}_1, ..., \mathbf{s}_k]^\top, \mathbf{X}), \mathbf{s}_j = PMA_1([\mathbf{s}_1, ..., \mathbf{s}_{j-1}]^\top) \text{ for } j = 2, ..., k, \tag{7}$$

which enables an RNN-like iterative computation by sequentially extending the parameters for PMA. The clustering network to output variable number of parameters is then defined as

$$\mathbf{H}_X = ISAB_L(\mathbf{X}), \mathbf{H}_\theta^{(k)} = SAB_{L'}(aPMA(\mathbf{H}, k)),$$

$$v_k = \text{SIGMOID}(\text{MEAN}(\text{ rFF}(\mathbf{H}_\theta^{(k)}[:, 1]))), \quad s_k = 1 - \prod_{j \le k} v_k \tag{8}$$

$$(\text{logit}\pi_j^{(k)}, \theta + j^{(k)})_{j=1}^k = \text{rFF}(\mathbf{H}_\theta^{(k)}[:, 2 :]),$$

where $[:, 1]$ and $[:, 2 :]$ are numpy-like notation indexing the columns. $s_k$ is a stop variable where $s_k > 0.5$ means the iteration stops at $k$-th step and continues otherwise.

During training, ST-ACT utilize the true number of clusters as supervision for training $c_t$, yielding the overall loss function

$$\mathbb{E}_{p(\mathbf{X}, k)} \left[ -\sum_{i=1}^N \log \sum_{j=1}^k \pi_j^k p(\mathbf{x}_i, \theta_j^{(k)}) + \sum_{t=1}^T BCE(c_t, \mathbb{1}_{\{t < K\}}) \right] \tag{9}$$

where $k$ is the true number of clusters, $T$ is maximum number of steps to run.

# B Model Analysis

## B.1 Relations to Neural (Gaussian) Processes

It is illustrative to compare the clustering factor $p(\mathcal{C}_k | \mathcal{C}_{1:k-1}, \mathcal{S}_k)$ with Gaussian Processes (GPs) and their representation via Neural Processes (NPs) [3]. The GPs predictive distribution, given context pairs $(\mathbf{x}_G, \mathbf{y}_G) = (\mathbf{x}_i, \mathbf{y}_i)_{i \in G}$ and $N$ targets $(\mathbf{x}_i, \mathbf{y}_i)_{i \in [1, N]}$, satisfies

$$\int p(\{\mathbf{y}_i\}_{i=1}^N | \{\mathbf{x}_i\}_{i=1}^N, \mathbf{x}_G, \mathbf{y}_G) \, d\mathbf{y}_1 = p(\{\mathbf{y}_i\}_{i=2}^N | \{\mathbf{x}_i\}_{i=2}^N, \mathbf{x}_G, \mathbf{y}_G) \tag{10}$$

i.e., the dependence on $\mathbf{x}_i$ disappears when we integrate it. Thus, unlike clustering models, there is no marginal persistency. The de Finetti representation of NPs for this distribution is

$$p\left(\{\mathbf{y}_i\}_{i=1}^N|\{\mathbf{x}_i\}_{i=1}^N, \mathbf{x}_G, \mathbf{y}_G\right) = \int \prod_{i=1}^N p\left(\mathbf{y}_i|\mathbf{x}_i, \mathbf{z}, \mathbf{x}_G, \mathbf{y}_G\right) p(\mathbf{z}|\mathbf{x}_G, \mathbf{y}_G)d\mathbf{z} \qquad (11)$$

and thus Eq. (10) is satisfied because the prior $p(\mathbf{z}|\mathbf{x}_G, \mathbf{y}_G)$ is independent of the target $\mathbf{x}_i$.

In the analogy with our MCP, the target pairs $(\mathbf{x}_i, \mathbf{y}_i)$ correspond to $(\mathbf{x}_i, q(b_{i,j}^{(k)}|\mathbf{x}_i))$ where $\mathbf{x}_i \in \mathcal{S}_k$, the context $(\mathbf{x}_G, \mathbf{y}_G)$ corresponds to the already clustered points in $\mathcal{C}_{1:k-1}$, and the de Finetti representation corresponds to following conditional distribution:

$$p(q_{i,j}^{(k)}|\mathcal{S}_k) \simeq \int df_\theta(\phi^{(k)}) \prod_{i=1}^{M_k} p\left(q_{i,j}^{(k)}|f_\theta(\phi^{(k)}), \mathbf{x}_i\right) p(f_\theta(\phi^{(k)})|\mathcal{S}_k) \qquad (12)$$

which is a hierarchical modeling of clustering generation by introducing latent cluster representation $\phi^{(k)}$. Here $q_{i,j}^{(k)}$ is $q(b_{i,j}^{(k)}|\mathbf{x}_i)$. However, the clustering prior $p(f_\theta(\phi^{(k)})|\mathcal{S}_k)$ depends also on the target point $\mathbf{x}_i$, thus leading $p(f_\theta(\phi^{(k)})|\mathcal{S}_k)$ to satisfy the following marginal persistence:

$$\sum_{q_{i,j}^{(k)} \in [0,1]} \sum_{j=1}^k p\left(q_{1,j}^{(k)}, \cdots, q_{M_k,j}^{(k)}|f_\theta(\phi^{(k)}), \mathcal{S}_k\right) = \sum_{j=1}^k p\left(q_{2,j}^{(k)}, \cdots, q_{M_k,j}^{(k)}|f_\theta(\phi^{(k)}), \mathcal{S}_k\right) \qquad (13)$$

which results from marginalizing any binary variable $q_{i,j}^{(k)}$ in $p(q_{i,j}^{(k)}|f_\theta(\phi^{(k)}), \mathcal{S}_k)$ still depends on the data point $\mathbf{x}_i$.

## B.2 Connect Mixture Model to Optimal Transport

Here we analysis the proposed *IntraCC* is equivalent to solving the entropic regularized Kantorovich relaxation of optimal transport introduced in Section A.1.

We consider a mixture model with $k$ components.

$$p(\mathbf{x}_i|\boldsymbol{\phi}) = \sum_{\mathbf{b}_i} p\left(\mathbf{x}_i, \mathbf{b}_i|\boldsymbol{\phi}\right) = \sum_{j=1}^k \underbrace{p(b_{i,j})}_{\pi_j} p\left(\mathbf{x}_i|\phi_j, b_{i,j}\right); \qquad (14)$$

where $\mathbf{b}_i = [b_{i,1}, b_{i,2}, \cdots, b_{i,k}]$ are latent variables encoding the data point assignments, such that $b_{i,j}$ indicates data point $i$ was generated by component $j$. $\boldsymbol{\pi} = [\pi_1, \pi_2, \cdots, \pi_k]$ are the mixing coefficients.

For simplicity, we set equal mixing proportions $1/k$ and fixed shared spherical covariances of Gaussian likelihood as $p(\mathbf{x}_i|\boldsymbol{\phi}) = \sum_{j=1}^k \frac{1}{k}\mathcal{N}(\mathbf{x}_i; \phi_j, \epsilon\mathbf{I})$. In this case, $\phi_j \in \mathbb{R}^d$ is the mean vector. In this case, $p(b_{i,j}) = 1/k$. Suppose data points $\mathbf{x}_i \in \mathcal{D}$ are i.i.d sampled, the log-likelihood of $\mathcal{D}$ under the model is $\log p(\mathcal{D}|\boldsymbol{\phi}) = \sum_{\mathcal{D}} \log p(\mathbf{x}_i|\boldsymbol{\phi})$.

We focus on optimizing a lower bound derived by the following Jensen variational lower bound:

$$\sum_{\mathcal{D}} \log p(\mathbf{x}_i|\boldsymbol{\phi}) = \sum_{\mathcal{D}} \log \sum_{j=1}^k p(\mathbf{x}_i, b_{i,j}|\boldsymbol{\phi}) \geq \sum_{\mathcal{D}} \sum_{j=1}^k q(b_{i,j}|\mathbf{x}_i) \log \frac{p\left(\mathbf{x}_i, b_{i,j}|\boldsymbol{\phi}\right)}{q(b_{i,j}|\mathbf{x}_i)} \qquad (15)$$

where $q(b_{i,j}|\mathbf{x}_i)$ is the variational distribution. Here we denote $q_{i,j}$ as $q(b_{i,j}|\mathbf{x_i})q(\mathbf{x_i})$ with $q(\mathbf{x_i}) = 1/N$. Then maximizing Eq.(15) with respect to the $\mathbf{Q} = (q_{i,j}) \in \mathbb{R}^{N \times k}$ is equivalent to

$$J(\mathbf{Q}) = \min_{\mathbf{Q}} \sum_{\mathcal{D}} \sum_{j=1}^k c_{i,j} q_{i,j} - \epsilon H(\mathbf{Q}) \qquad (16)$$

where $c_{i,j} = -\log p\left(\mathbf{x}_i, b_{i,j}|\boldsymbol{\phi}\right) = \frac{||\mathbf{x}_i - \phi_j||^2}{2\epsilon} + \text{const}$. By having the quadratic kernel $\mathcal{K}(x, \phi) = -\frac{1}{2}||x - \phi||^2$, we can see Eq.(16) has strong connection to Eq.(1), i.e., Mixture-based OT loss Eq.(16) reduces to original OT loss Eq.(16) except for the constraints $\sum_{i=1}^N q_{i,j} = \frac{1}{k}$, $\forall j$ (Note that the constraint $\sum_{j=1}^k q_{i,j} = \frac{1}{N}$, $\forall i$ is automatically met).

## B.3 Connect Induced Self-Attention Block (ISAB) to OT-based Mixture Model

First, we give the additional implementation details of self attention module. The whole training procedure related to each component is also given in this section. Before that, multi-head attention between two different sets is defined as [7]

$$MAB(\mathbf{X}, \mathbf{Y}) = \mathbf{H} + rFF(\mathbf{H}), \quad \mathbf{H} = \mathbf{X} + rFF(MhA(\mathbf{X}, \mathbf{Y})) \tag{17}$$

where $rFF()$ is a feed-forward layer applied row-wise (i.e., for each element). We use Multi-head attention Module $\text{MHA}(\cdot)$ to exploit pair-wise or higher-order interactions between data points in both inter-and intra-cluster. Considering we want to capture the elements-wise relationship between $\mathbf{X}$ and $\mathbf{Y}$, we set $\mathbf{X}$ as query, and set key and values are $\mathbf{Y}$. The Multi-head attention Module is defined as follow

$$\text{MHA}(\mathbf{X}, \mathbf{Y}) = \text{CONCAT}(\mathbf{O}_1, \cdots, \mathbf{O}_H)\mathbf{W}^h, \quad \mathbf{O}_i = \sigma\left(\mathbf{X}\mathbf{W}_i^Q \left(\mathbf{Y}\mathbf{O}_i^K\right)^\top\right)\mathbf{Y}\mathbf{W}_i^V \tag{18}$$

where $\sigma(\cdot)$ is activation function, $\mathbf{W}_i^Q, \mathbf{W}_i^K, \mathbf{W}_i^V$ are head-specific transform matrices.

A Self-Attention Block (SAB) is simply MAB applied to the set itself: $SAB(\mathbf{X}) \triangleq MAB(\mathbf{X}, \mathbf{X})$. Note that the time-complexity of SAB is $O(N^2 d)$ because of pairwise computation. To reduce this, Lee et al. [7] proposed to use Induced Self-Attention Block (ISAB) defined as

$$ISAB(\mathbf{X}) = MAB(\mathbf{X}, MAB(\phi, \mathbf{X})), \tag{19}$$

where $\phi \in [\phi_1, \phi_2, \cdots, \phi_k]^\top \in \mathbb{R}^{k \times d}$ are trainable inducing points. ISAB indirectly compares the elements of $\mathbf{X}$ through the inducing points, reducing the time-complexity to $O(Nk)$.

Actually, ISAB is equivalent our *IntraCM* [5, 6]. Here we give a detailed analysis. A matrix form of ISAB is given as follows:

$$\phi_j \leftarrow \sum_i \frac{\exp((\mathbf{W}^Q\phi_j)^\top(\mathbf{W}\mathbf{x}_i)/\sqrt{d})}{\sum_i \exp((\mathbf{W}^Q\phi_j)^\top(\mathbf{W}\mathbf{x}_i)/\sqrt{d})}\mathbf{W}\mathbf{x}_i \tag{20}$$

where $\mathbf{W}^Q$ and $\mathbf{W}$ are the weight matrices for queries and keys/values, respectively. Considering $\mathbf{W}^Q$ and $\phi$ are both learnable parameters, we can get a more general ISAB as follow:

$$\phi_j \leftarrow \sum_i \frac{\exp(\phi_j^\top(\mathbf{W}\mathbf{x}_i)/\sqrt{d})}{\sum_i \exp(\phi_j^\top(\mathbf{W}\mathbf{x}_i)/\sqrt{d})}\mathbf{W}\mathbf{x}_i \tag{21}$$

We consider the optimal transport problem between $\phi$ and $\mathbf{X}$ only with constraint $\sum_{i=1}^N q_{i,j} = \frac{1}{k}, \forall j$, i.e.,

$$J(\mathbf{Q}) = \min_{\mathbf{Q}} \sum_{\mathcal{D}} \sum_{j=1}^k c_{i,j} q_{i,j} - \epsilon H(\mathbf{Q}), \quad \text{s.t.} \quad \sum_{i=1}^N q_{i,j} = \frac{1}{k}, \forall j \tag{22}$$

where $c_{i,j} = -\mathcal{K}(\mathbf{x}_i, \phi_j) = -\phi_j^\top \mathbf{W}\mathbf{x}_i$ corresponds to a linear kernel. We can solve it via Lagrange multiplier and get

$$q_{i,j} = \exp(-c_{i,j}/\epsilon + \alpha_j) \tag{23}$$

where $\alpha_j$ is a constant. By combining the constraint $\sum_i q_{i,j} = 1/k$, we can get

$$\exp(\alpha_j) = \frac{1}{k \sum_i \exp(-c_{i,j}/\epsilon)}, \quad q_{i,j} = \frac{1}{k}\frac{\exp(\phi_j^\top(\mathbf{W}\mathbf{x}_i)/\epsilon)}{\sum_i \exp(\phi_j^\top(\mathbf{W}\mathbf{x}_i)/\epsilon)} \tag{24}$$

By setting $\epsilon = \sqrt{d}$, we can get the updated $\phi_j$ as follows:

$$\phi_j \leftarrow \sum_i \frac{\exp(\phi_j^\top(\mathbf{W}\mathbf{x}_i)/\sqrt{d})}{\sum_i \exp(\phi_j^\top(\mathbf{W}\mathbf{x}_i)/\sqrt{d})}\mathbf{W}\mathbf{x}_i \tag{25}$$

Note that here we set a linear kernel to analysis the relation between the proposed IntraCM and ISAB, we give a more general form (non-linear neural network-based kernel) in main manuscript which is able to achieve more flexible solution.

### B.4 Connection to Existing Amortized Clustering

Different traditional clustering relies on point-wise learning manner, amortized clustering (e.g., ST-ACT, DAC, NCP, CHiGac and our AMCP) allows cluster-wise learning strategy (learning one cluster one time), which is more efficient. However, ST-ACT inherits the feature of MoG which needs to manually set the number of clusters. DAC generalizes ST-ACT and enables to produce a varying number of clusters. Both ST-ACT and DAC can be regarded as simply amortized inference of Gaussian mixture with the aid of neural networks. The attention mechanism used in ST-ACT and DAC are much complexed than the proposed optimal transport mixing reducing the quadratic cost of traditional similarity weight computation. The most similar work to our methods is NCP [10] and CHiGac [9]. Both of them explain clustering from the perspective of generative models, which similar in spirit to the popular Gibbs sampling algorithm for Dirichlet process mixture models, but without positing particular priors on partitions. Besides, NCP and CHiGac rely heavily on the selecting of anchor points and the processing ordering and often exhibits unstable properties. Different from previous work, AMCP connects the relationship of mixture model and optimal transport to describe the cluster's summary statistics well, and seamlessly combines mixing and coupling to achieve efficient amortized clustering.

### B.5 Complexity Analysis

For our AMCP, *IntraCM* perform $T$ EM steps, each E-step amounts to computing the Gaussian likelihoods, and additionally running of Sinkhorn's algorithm if we adopt OT-based E step. M-step mainly consists of the weighted sum of partial derivative. Thus the complexity of regular E-step takes $O(Nkd)$, and $O(Nk(L+d))$ for OT-based E step where $L$ is the number of iterations of Sinkhorn's algorithm. The complexity of M-step takes $O(Nkd)$. Thence, the time complexity for EM-step is $O(Nkd)$ or $O(Nk(d+L))$. Usually, $L$ is set to $5-10$, which is largely smaller than $d$.

We know that $k$ is largely smaller than $N$, which indicates $O(Nkd)$ or $O(Nk(d+L))$ is largely lower than $O(N^2 d)$ used in attention-based methods (e.g., DAC, ST-ACT). Although DAC and ST-ACT introduce induced self-attention block which is able to achieve complexity $O(Nkd)$ (for fair comparison, here we set the number of trainable inducing points to $k$), they cannot avoid computing the traditional attention module with complexity $O(N^2)$.

## C Experimental Details and Additional Results

### C.1 Infrastructure and Experimental Details

**Infrastructure:** We implement our model with Pytorch, and conduct our experiments with:

- CPU: Intel Xeon Silver 4116 @2.1GHz.
- GPU: 8x GeForce RTX 2080Ti.
- RAM: DDR4 256GB.
- ROM: 8x 1TB 7.2K 6Gb SATA and 1x 960G SATA 6Gb R SSD.
- Operating system: Ubuntu 18.04 LTS.
- Environments: Python 3.7; NumPy 1.18.1; SciPy 1.2.1; scikit-learn 0.23.2; seabornn 0.1; torch_geometric 1.6.1; matplotlib 3.1.3; dgl 0.4.2; pytorch 1.6.

**Hyper-parameter search:** We trained with the following hyperparameters: The neural network (e.g., $f_\theta(\cdot)$) in our model is a multilayer perceptron (MLP). We use the *tanh* activation function. We apply dropout before every layers, except the last layer. The model is trained using Adam. We then tune the other hyper-parameters of both our approachs and our baselines automatically using the TPE [1] method implemented by Hyperopt [2]. We let Hyperopt conduct 200 trials to search for the optimal hyper-parameter configuration for each method on the validation of each dataset. The hyper-parameter search space is specified as follows:

- The number of hidden layers in a neural network: {0, 1, 2, 3 }.
- The number of neurons in a hidden layer: { 100, 200, $\cdots$, 1000 }.

Table 1: Summary of experimental real-world datasets.

| Dataset | MNIST | Tiny-ImageNet | CIFAR-10 |
|---|---|---|---|
| ♯ Samples | 70,000 | 110,000 | 60,000 |
| ♯ Input Dimensions | 784 | 4,096 | 1,024 |
| ♯ Clusters | 10 | 200 | 10 |

- Learning rate: $[10^{-8}, \cdots, 1]$.
- L2 regularization: $[10^{-12}, \cdots, 1]$.
- Dropout rate: $[0.05, \cdots, 1]$.
- Regularization coefficient $\lambda$: $[1, 10]$.
- The standard deviation of the prior for $\{\theta\}$: $[0.01, 0.5]$.

**Datasets:** Three real-world datasets (*MNIST, Tiny-ImageNet, CIFAR-10*) are used to evaluate the proposed model.

- *MNIST*: The MNIST dataset consists of 70,000 hand-written digits. The images are centered and of size $28 \times 28$ pixels. We reshaped each image to a 784-dimensional vector.
- *Tiny-ImageNet*: Tiny-ImageNet contains 100,000 images of 200 classes (500 for each class) downsized to $64 \times 64$ colored images chosen from ImageNet ILSVRC-2012 challenge. Each class has 500 training images, 50 validation images and 50 test images.
- *CIFAR-10*: The CIFAR-10 dataset (Canadian Institute for Advanced Research, 10 classes) is a subset of the Tiny Images dataset and consists of 60,000 $32 \times 32$ color images. The images are labelled with one of 10 mutually exclusive classes: airplane, automobile (but not truck or pickup truck), bird, cat, deer, dog, frog, horse, ship, and truck (but not pickup truck). There are 6,000 images per class with 5000 training and 1000 testing images per class.

The overall statistical information about datasets is given in Table 1.

Note that both *MNIST* and *CIFAR-10* have 10 classes and Tiny-ImageNet have 200 classes. To formulate amortized mechanism, we process the dataset as follow:

- For *MNIST* and *CIFAR-10*, we define the data containing the first 5 classes as the training set, and the last 5 classes as the test set. Specifically, for *MNIST*, training set contains samples related to digits 0, 2, 4, 6, 8, and test set is related to 1, 3, 5, 7, 9. For *CIFAR-10*, training set contains samples related to airplane, bird, deer, frog, ship, and the rest is the test set.
- For *Tiny-ImageNet*, we define the data containing the first 100 classes as the training set, and the remaining 100 classes as the test set (the class order refers to the order in which it appeared in the original *Tiny-ImageNet*).

We sample multiple clustered datasets to form training sets with $N \in \{1000, 10000\}$. For test set, we generate 1000 randomly clustered datasets with $N \in \{1000, 5000\}$.

**Evaluation Metrics:** Two type of evaluation metrics are adopted. Clustering accuracy

$$ACC = \max_{m \in \mathcal{M}} \frac{1}{N} \mathbb{1}\{y_i = m(\hat{y}(\mathbf{x}_i))\}$$

and Normalized Mutual Information

$$NMI(\mathcal{C}_i, \mathcal{C}_j) = \frac{MI(\mathcal{C}_i, \mathcal{C}_j)}{\sqrt{H(\mathcal{C}_i)H(\mathcal{C}_j)}},$$

where $N$ is the total number of data samples, $y_i$ is the ground-truth label that corresponds to that $\mathbf{x}_i$ sample, $\hat{y}(\mathbf{x}_i)$ is the cluster assignment obtained by the model, and $m$ ranges over the set $\mathcal{M}$ of all possible one-to-one mappings between cluster assignments and labels. Larger *ACC* value indicates better performance. For *NMI*, $MI(\cdot, \cdot)$ denotes the mutual information and $H(\cdot)$ is the entropy. This measure can be applied to any assignment coming from the clusters or the true labels. If the two

Table 2: A toy clustering results comparisons among different amortized clustering methods (DAC, NCP, ST-ACT and AMCP) on *MNIST* dataset.

Table 3: A toy clustering results comparisons among different amortized clustering methods (DAC, NCP, ST-ACT and AMCP) on *Tiny-ImageNet* dataset.

assignments $\mathcal{C}_k$ and $\mathcal{C}_j$ are independent, the *NMI* is equal to 0. If one of them is deterministically predictable from the other, the *NMI* is equal to 1.

**Model Architecture:** Here, we give the architectural details of the proposed AMCP models used for the synthetic and real-world experiments. We implement $f_\theta(\cdot)$ by means of a multilayer perceptron (MLP) with a sigmoid output. The representation $\phi_j^{(k)}$ is a real-value $d$-dimensional vector squashed to the $(0, 1)$ range by a sigmoid function before being fed into the networks. For synthetic data, we set $d = 2$. For real-world datasets, we set $d = 512$.

For DAC, NCP and ST-ACT, we used the code released by the authors with default hyper-parameters. For ST-ACT and DAC, the network architecture is $ISAB(m, dh, 4) \rightarrow ISAB(m, dh, 4) \rightarrow PMA(4, dh, 4) \rightarrow SAB(dh, 4) - SAB(dh, 4) - FC(d_{out})$ where $SAB(d, h)$ means the set attention block with $d/h$ units/heads, and $PMA(d, h, k)$ is the pooling multihead attention layer with $k$ vectors and $d/h$ units/heads. For NCF, there are composed by the deep networks $h$, $g$, $u$, $f$. For 2D Gaussians, $h$ is $MLP[2 - 256 - 256 - 256 - 128]$ with ReLUs; $u$ is $MLP[2 - 256 - 256 - 256 - 128]$ with ReLUs; $g$ is $MLP[128 - 256 - 256 - 256 - 256]$ with ReLUs; $f$ is $MLP[384 - 256 - 256 - 256 - 1]$ with ReLUs. For real-world datasets, $h$ is 2 layers of $[convolutional + maxpool + ReLU] + MLP[320 - 256 - 128]$ with ReLUs; $u$ is same as $h$; $g$ is $MLP[256 - 128 - 128 - 128 - 128 - 256]$ with ReLUs; $f$ is $MLP[384 - 256 - 256 - 256 - 1]$ with ReLUs.

For all baselines, we use ADAM as optimizer with learning rate $10^{-4}$ to train models..

## C.2 Additional Experimental Results

### C.2.1 Cluster Generative Process

To illustrate how the clustering models capture the shape ambiguity of some of the images, we plot the ground-truth image clusters and clustering results (obtained by DAC, NCP, ST-ACT and AMCP respectively) on *MNIST*, *Tiny-ImageNet* and *CIFAR-10*, as shown in Table 2, 3 and 4. Comparing with DAC, NCP, ST-ACT, the proposed AMCP can get the most accurate results. Taking *MNIST* dataset as an example, DAC assigns the digit 7 (with similar appearance to 9) to cluster 9, and NCP generates a new cluster for it. Fortunately, ST-ACT and DAC correctly assigns it to cluster 7. All baselines are unable to assign the digit 4 written in a strange way to the right cluster, and our AMCP correctly assigns it to cluster 4. Similar results can be found in other two datasets.

### C.2.2 Convergence

We investigate the evolution of the model performance along training epochs and test epochs. Figure 1(a) and (b) shows the training and test likelihood vs. epoch for NCP and our AMCP on synthetic data. Figure 1(c) and (d) shows the training and test accuracy vs. epoch for NCP and our AMCP on *CIFAR-10*. We can see that AMCP can achieve better performance than NCP on both synthetic and real-world datasets, and AMCP converges faster than NCP.

Table 4: A toy clustering results comparisons among different amortized clustering methods (DAC, NCP, ST-ACT and AMCP) on *CIFAR-10* dataset.

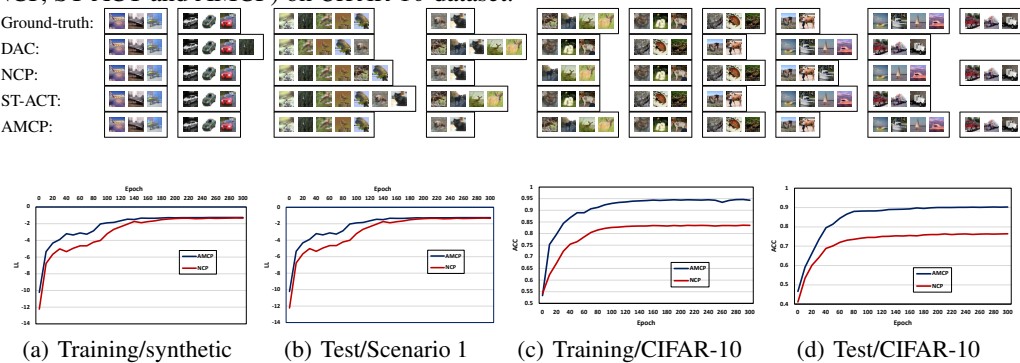

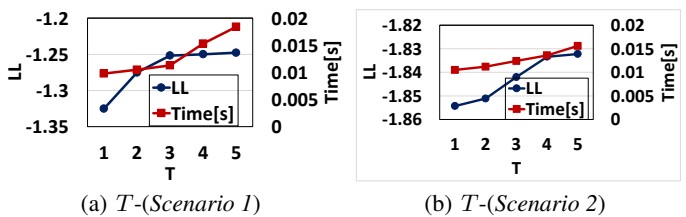

(a) Training/synthetic  (b) Test/Scenario 1  (c) Training/CIFAR-10  (d) Test/CIFAR-10

Figure 1: Training and test curves for the AMCP and NCP of synthetic data (Scenario 1) and *CIFAR-10*.

### C.2.3  Empirical Study on Hyper-parameters Impacts

We empirically analyze the proposed model from the impact of various hyper-parameters. They are summarized as follows.

**The Number of EM Steps $T$:**  AMCP can apply multiple EM steps to achieve more accurate $\phi^{(K,T)}$. Here we conduct experiments on varying $T$ in $\{1, 2, 3, 4, 5\}$ to investigate the effect of $T$. Figure 2 and 3 show the effect of $T$ on synthetic data (*Scenario 1* and *Scenario 2*) and real-world data (*MNIST*, *Tiny-ImageNet*, *CIFAR-10*). We can see that, the performance (*LL* and *ACC*) become better and better with the increasing of $T$, while the running time is linearly scalable to the number of steps.

**OT E-step vs. Regular E-step** We introduce two different ways to conduct E-step in IntraCM: regular soft-assignment E-step and Sinkhorn-based optimal transport E-step. Figure 4 and 5 show the comparison between regular E-step and OT E-step on synthetic data (*Scenario 1* and *Scenario 2*) and real-world data (*MNIST*, *Tiny-ImageNet*, *CIFAR-10*). The maximum number of Sinkhorn-Knopp iterations for solving the OT problem is set to 10. While the regular E-step is consistently faster than the OT E-step since it's closed-form solution, OT E-step can achieve more accurate likelihood and accuracy.

**Update Previous Clusters** In order to prevent the previous generated cluster from inaccurate, we can update previous clusters as stated in line 12 of Algorithm 1. Here we investigate the pros and cons of the two strategies by comparing whether previous clusters are updated or not, as shown in Table 5 and 6. We can see that AMCP* can achieve more better performance in more complicated scenario (*Scenario 2*, *Tiny-ImageNet* and *CIFAR-10*). The possible reason is that a single step to form cluster cannot exploiting data relations well, whole the update strategy is able to update previous clusters iteratively.

**The Number of Classes in Training set** Here we conducts experiments to investigate the model performance on different size of training set. Figure 6 shows the performance comparison on different training set with different number of classes. For *MNIST* and *CIFAR-10*, the number of classes in training set is turned from 1 to 5. For *Tiny-ImageNet*, the number of classes is turned in $\{20, 40,$

(a) $T$-(*Scenario 1*)  (b) $T$-(*Scenario 2*)

Figure 2: The impact of $T$ on synthetic datasets.

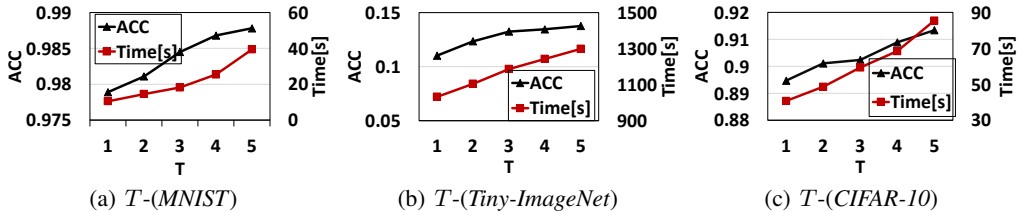

(a) $T$-(*MNIST*)          (b) $T$-(*Tiny-ImageNet*)          (c) $T$-(*CIFAR-10*)

Figure 3: The impact of $T$ on real-world datasets.

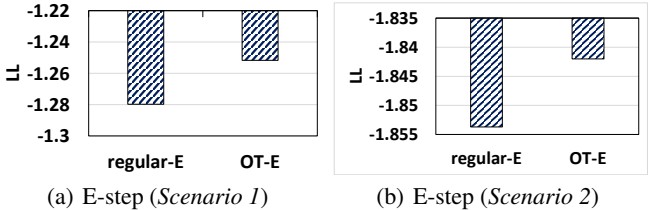

(a) E-step (*Scenario 1*)          (b) E-step (*Scenario 2*)

Figure 4: OT E-step vs regular E-step on synthetic datasets.

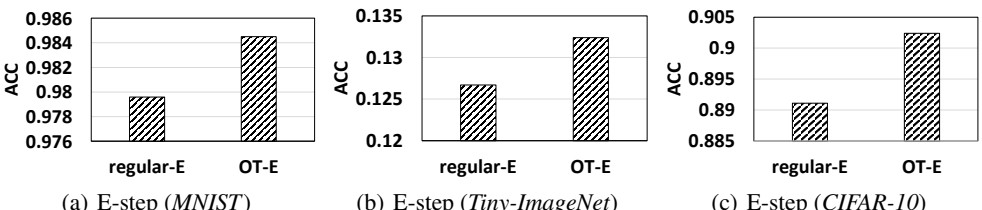

(a) E-step (*MNIST*)          (b) E-step (*Tiny-ImageNet*)          (c) E-step (*CIFAR-10*)

Figure 5: OT E-step vs regular E-step on real-world datasets.

Table 5: Comparing where previous clusters are updated or not in synthetic datasets. Here AMCP* indicates methods updating previous clusters.

| Dataset | Scenario 1 | | Scenario 2 | |
|---|---|---|---|---|
| Metric | *LL* | Time[s] | *LL* | Time[s] |
| AMCP | **-1.2517**$_{\pm 0.035}$ | **0.0113**$_{\pm 0.003}$ | -1.8420$_{\pm 0.025}$ | **0.0124**$_{\pm 0.002}$ |
| AMCP* | -1.2587$_{\pm 0.052}$ | 0.0143$_{\pm 0.003}$ | **-1.8323**$_{\pm 0.056}$ | 0.0154$_{\pm 0.003}$ |

Table 6: Comparing where previous clusters are updated or not in real-world datasets. Here AMCP* indicates methods updating previous clusters.

| Dataset | *MNIST* | | *Tiny-ImageNet* | | *CIFAR-10* | |
|---|---|---|---|---|---|---|
| Metric | *ACC* | Time[s] | *ACC* | Time[s] | *ACC* | Time[s] |
| AMCP | **0.9845**$_{\pm 0.001}$ | **18.22**$_{\pm 3.74}$ | 0.1324$_{\pm 0.002}$ | **1186.4**$_{\pm 19.37}$ | 0.9024$_{\pm 0.002}$ | **59.33**$_{\pm 5.59}$ |
| AMCP* | 0.9764$_{\pm 0.002}$ | 21.44$_{\pm 2.85}$ | **0.1378**$_{\pm 0.002}$ | 1273.4$_{\pm 26.25}$ | **0.9085**$_{\pm 0.003}$ | 65.44$_{\pm 4.21}$ |

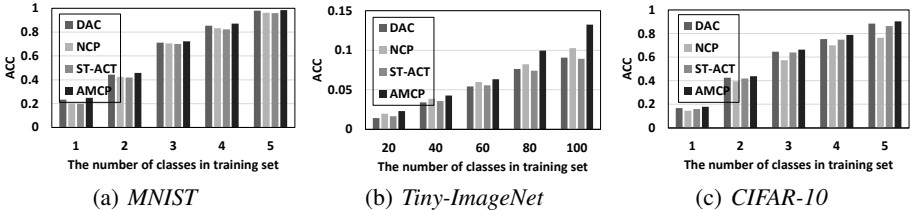

(a) *MNIST*     (b) *Tiny-ImageNet*     (c) *CIFAR-10*

Figure 6: Performance (ACC) comparison on different training set with different number of classes.

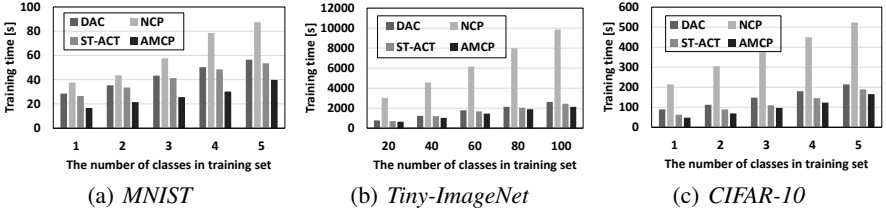

(a) *MNIST*     (b) *Tiny-ImageNet*     (c) *CIFAR-10*

Figure 7: Training time on different training sets with different number of classes.

60, 80, 100}. We can see the accuracy of all methods increases with the number of classes, because more classes in the training data help the test data to scale quickly. AMCP can achieve optimal results in all training scenarios, which proves the effectiveness of AMCP. Further, we report the trining time in Figure 7. We can see that AMCP still outperforms baselines.