# OpenReview forum: "Amortized Mixing Coupling Processes for Clustering"
_NeurIPS.cc/2022/Conference — NeurIPS 2022 Accept_

### Official Review · Reviewer_XhPA · 2022-07-05

**Rating:** 6
**Confidence:** 4
**Soundness:** 4 excellent
**Presentation:** 3 good
**Contribution:** 3 good

**Summary:**

This paper proposes a novel amortized clustering framework coupled with two key strategies named Intra-Cluster Mixing and Inter-Cluster Coupling. Intra-Cluster Mixing utilizes optimal transport to learn the plan for distributing the mass of data to differential reference vectors. Inter-Cluster then resorts to this plan to form the next cluster and update existing clusters. According to the results on both synthetic and real-world datasets, the proposed method enjoys superior performance and computational efficiency.

**Questions:**

The settings of prior distribution are directly given in experiments, e.g., $l_{1,K}\\sim U(0,9)$ in line 302. I wonder whether the performance is sensitive to the choice of prior.

**Limitations:**

Limitations of this work are presented in the conclusion. This work is a technical clustering approach and has few social impacts.

**Strengths And Weaknesses:**

Strengths:
The proposed method is novel and practical. In contrast to previous point-wise amortized clustering methods, the proposed framework directly models the cluster generative process. Introducing optimal transport in the E step to optimize the ELBO of mixing likelihood is also a highlight. Besides, this paper is well-organized and easy to follow. The proposed method is technically sound.

Weaknesses: The explanation of Figure 2 seems too concise. It would be better if more details are provided to make the proposed learning procedure more understandable. Besides, the difference between the proposed method and the literature [23] is not clear. The authors are encouraged to provide more details on the differences.

---

> ### Author Response · Authors · 2022-08-01
> **Response to Reviewer XhPA**
>
> Thanks very much for your valuable and comprehensive comments!
>
> **Q1. The explanation of Figure 2 seems too concise. Difference from the existing work.**
>
> Thanks for your suggestion, we add more detailed explanation of Figure 2 (Figure 1 (b) and (c) in the revised version). The proposed cluster-wise AMCP is compose of two key strategies: Intra-cluster mixing (IntraCM) and inter-cluster coupling (InterCC). IntraCM and InterCC performs to generate cluster one by one. Among them, IntraCM cooperates optimal transport from mixture view and describe the cluster's summary statistics with the aid of multiple EM steps. Then, InterCC regards the output of IntraCM as data coupling to generate the next cluster.
>
> For the differences between our method and related work, we have reorganized the related works, and clear how the proposed method differs from the existing work in the revised version. Specifically, among existing amortized clustering models, ST-ACT inherits the feature of GMM which needs to manually set the number of clusters. DAC generalizes ST-ACT and enables to produce a varying number of clusters. Both ST-ACT and DAC can be regarded as simply amortized inference of Gaussian mixture with the aid of neural networks. The attention mechanism used in ST-ACT and DAC are much complexed than the proposed optimal transport mixing reducing the quadratic cost of traditional similarity weight computation. The most similar work to our methods is NCP and CHiGac. Both of them explain clustering from the perspective of generative models, which similar in spirit to the popular Gibbs sampling algorithm for Dirichlet process mixture models, but without positing particular priors on partitions. Besides, NCP and CHiGac rely heavily on the selecting of anchor points and the processing ordering and often exhibits unstable properties. Different from previous work, AMCP connects the relationship of mixture model and optimal transport to describe the cluster's summary statistics well, and seamlessly combines mixing and coupling to achieve efficient amortized clustering.
>
> More detailed analyses are also given in revised Supplementary Materials (B.4).
>
> **Q2. Whether the performance is sensitive to the choice of prior in generative process.**
>
> The parameter of prior distribution indicates the number of classes in the dataset we want to generate. As we know, both MNIST and CIFAR-10 have 10 classes and Tiny-ImageNet have 200 classes. To formulate the amortized mechanism, for all dataset, data containing half of the classes is used to generate the training set, and the remaining half is used to generate the test set, with no overlap between training classes and test classes. To better understand the generative process of training and testing data, we reformulate the generative process given in the main manuscript, i.e.,
>
> $\alpha \sim  Exp(1)$, $\ c_{1:N} \sim CRP(\alpha)$, $K-1 \sim  Binomial(K_g -1, 0.5)$, $l_{k} \sim U(0,K-1)$, $x_i \sim U[D, l_{c_i}]$.
>
> Here $x_i \sim U[D, l_{c_i}]$ indicates sampling $x_i$ with label $l_{c_i}$ uniformly from dataset $D$, which can be training set or testing set. $K_g$ indicates the number of clusters we can sample, $K$ is sampled from Binomial distribution with parameter $K_g$ and 0.5. In this case, the parameter $K$ indicates the number of classes we are able to sample. More detailed data set information can be found in Supplementary Material (C.1)
>
> In this case, we conducted experiments by sampling training datasets with different $K$ (the number of classes) and evaluate the model performance. We compare different amortized clustering models under different training sets (contains 1-5 classes respectively) on MNIST dataset. Note that we define the test set contains all test classes (the remaining 5 classes). As shown in the following table, we list accuracy comparison of different methods under different $K$. We can see that the accuracy of all methods increases with the number of classes, because more classes in the training data help the test data to scale quickly. AMCP can achieve optimal results in all training scenarios, which proves the effectiveness of AMCP. More comprehensive experimental results and analysis on all datasets are given in Supplementary Material (C.2.5).
>
> |Methods |	1|	2|	3|	4|	5|
> |  ----  | :----:  | ----  | ----  | ----  |----  |
> |DAC	|0.2341|	0.4425|	0.7125|	0.8543|	0.9796|
> |NCP|	0.2015|	0.4236|	0.7052	|0.8325|	0.9633|
> |ST-ACT	|0.1989|	0.4201	|0.7011	|0.8235|	0.9596|
> |AMCP|	0.2488	|0.4569|	0.7226	|0.8721	|0.9845|
>
> Finally, thanks again for your valuable comments.

---

> > ### Comment · Reviewer_XhPA · 2022-08-09
> > **Further Comment**
> >
> > I would like to appreciate the prompt and rewarding responses given by the authors.

---

> > > ### Author Response · Authors · 2022-08-09
> > > **Thanks for your further comment**
> > >
> > > Thank you very much for your positive feedback. Our paper won't be better without your nice suggestions. Thanks again.

---

### Official Review · Reviewer_9BaR · 2022-07-11

**Rating:** 6
**Confidence:** 4
**Soundness:** 3 good
**Presentation:** 3 good
**Contribution:** 2 fair

**Summary:**

The paper proposes an amortised clustering process model, where the clustering prior is learnt in a meta-learning setting.

Inference is performed cluster-wise sequentially, involving inter- and intra-cluster similarities. To avoid calculations involving distances between all pairs of clusters, the authors use a linear optimal transport strategy.

The method is demonstrated on synthetic gaussian mixture models as well as meta-learnt clustering in MNIST, Tiny-ImageNet and CIFAR-10.

**Questions:**

* In section 4.1, I am confused by the statement "In this case, the training set contains 200 samples belonging to 4 clusters". I thought that amortised clustering methods would be trained on multiple datasets by repeated sampling from the CRP prior, with a variable number of clusters. Could you please clarify what is this "training set" refers to?

* Are results in Tables 1 and 2 based on a single test set case, or averaged over multiple (potentially varying-sized) test sets? For example, for MNIST visualisation, a subset is used, but what about the results in Table 2?

Minor:

* The results show that the proposed method outperforms other amortised methods at test time. It could be interesting to see whether this is the case throughout training (e.g. how the training/test curves look like as a function of iterations)?

* Right now, experiments are only presented as summary statistics. While this is common, it could be more insightful to see a qualitative toy example, e.g. some visualisation where the proposed OT-based methodology behaves differently to an existing model (say NCP).

**Limitations:**

No concerns

**Strengths And Weaknesses:**

The paper is well-written and quite clear, though some parts (sections 3.2.1 and 3.2.2) are quite dense with equations. Minor suggestion: For easier readability, it might make sense to merge Figures 1 and 2.

The idea to make use of optimal transport within the inference scheme is interesting and novel, and the authors demonstrate that it's beneficial over the standard EM algorithm. I also liked how the connection between their OT-based approach and ISAB was outlined in the appendix.

The authors emphasise that their approach is sequential per cluster and not per data point. While true, it seems to me that a clusterwise approach has been previously presented in [Pakman et al, 2020]?

I believe the significance of this type of work will remain relatively limited (In fact, I haven't seen _any_ of the existing work in amortised clustering processes being used for any non-toy tasks yet. On the other hand, one could argue that developing better models/inference algorithms can have the potential to change it -- in that case, ideally, it would be desirable to see convincing qualitative improvements over previous results, or a non-toy application example).

I have some questions about the experiments, I have outlined them below.

---

> ### Author Response · Authors · 2022-08-01
> **Response to Reviewer 9BaR**
>
> Thanks very much for your valuable and comprehensive comments!
>
> **Q1. For easier readability, it might make sense to merge Figures 1 and 2.**
>
> We have merged Figure 1 and 2 in the revised version.
>
> **Q2. Difference to existing amortized clustering**
>
> Among existing amortized clustering models, ST-ACT inherits the feature of GMM which needs to manually set the number of clusters. DAC generalizes ST-ACT and enables to produce a varying number of clusters. Both ST-ACT and DAC can be regarded as simply amortized inference of Gaussian mixture with the aid of neural networks. The attention mechanism used in ST-ACT and DAC are much complexed than the proposed optimal transport mixing reducing the quadratic cost of traditional similarity weight computation. The most similar work to our methods is NCP and CHiGac. Both of them explain clustering from the perspective of generative models, which similar in spirit to the popular Gibbs sampling algorithm for Dirichlet process mixture models, but without positing particular priors on partitions. Besides, NCP and CHiGac rely heavily on the selecting of anchor points and the processing ordering and often exhibits unstable properties. Different from previous work, AMCP connects the relationship of mixture model and optimal transport to describe the cluster's summary statistics well, and seamlessly combines mixing and coupling to achieve efficient amortized clustering.
>
> We have reorganized the related work, and clear how the proposed method differs from the existing work in the revised version. More detailed analyses are also given in revised Supplementary Materials (section B.4).
>
> **Q3. The significance of this type of work will remain relatively limited.**
>
> I agree with you that this type of work has limited results right now. I think the work of amortized clustering makes sense, especially when we are faced with classes of data that we have never seen before. That's why we focus on this type of work. In our work, our experiments on synthetic and real-world datasets (MNIST, Tiny-ImageNet, and CIFAR-10) hope to verify the effectiveness of the model (the ability of generalization to unseen classes).
>
>
> **Q4. Could you please clarify what is this "training set" refers to in section 4.1?**
>
> At each training step, we generate 10 random datasets according to the generative process. Each dataset contains 200 points on a 2D plane, each sampled from one of 4 Gaussians. Thank you for pointing out our lack of detail, we have described it in more detail in the revised version.
>
> **Q5. The results in Tables 1 and 2.**
>
> In Table 1 and 2, the average on 5 runs are reported. For MNIST toy-clustering results in Table 3 only one random selected testing result for each method is visualized. We have cleared this point in the revised version.
>
> **Q6. It could be interesting to see whether this is the case throughout training.**
>
> Thanks for your suggestion, please forgive us for not being able to show pictures in rebuttal’s reply box, and we give the training and test performance vs. epoch for NCP and our AMCP on synthetic data and CIFAR-10 dataset in the Supplementary Material (C.2.3). AMCP can achieve better performance than NCP, and AMCP converges faster than NCP.
>
> **Q7. It could be more insightful to see a qualitative toy example.**
>
> To illustrate how the clustering models capture the shape ambiguity of some of the images, we visualize the ground-truth image clusters and clustering results (obtained by DAC, NCP, ST-ACT and AMCP respectively) on MNIST in the main manuscript, as shown in Table 3. In original Supplementary Material (C.2.1), we give full experimental results of other two datasets (Tiny-ImageNet and CIFAR-10). In MNIST, DAC assigns the digit 7 (with similar appearance to 9) to cluster 9, and NCP generates a new cluster for it. Fortunately, ST-ACT and DAC correctly assign it to cluster 7. All baselines are unable to assign the digit 4 written in a strange way to the right cluster, and our AMCP correctly assigns it to cluster 4. Similar results can be found in other two datasets.
>
> To further investigate the model efficiency, we compare AMCP with NCP by the histograms of log-likelihood. The histograms of the log-likelihood per test example are presented in Supplementary Material (C.2.4). We notice that all histograms characterize a heavy-tail indicating existence of examples that are hard to represent. However, taking a closer look at the histograms for AMCP reveals that there are less hard examples comparing to the NCP, which indicates the proposed optimal transport mixing is beneficial for exploiting pair-wise interaction and further improving clustering performance.
>
> Finally, thanks again for your valuable comments.

---

> > ### Comment · Reviewer_9BaR · 2022-08-07
> > **Thank you for the response**
> >
> > Thank you for your thorough response, and for updating the paper with various clarifications. I have now updated my score to recommend acceptance.

---

> > > ### Author Response · Authors · 2022-08-09
> > > **Thank you for increasing score**
> > >
> > > Thank you very much for increasing score! Our paper won't be better without your valuable suggestions. Thanks again.

---

### Official Review · Reviewer_qZcB · 2022-07-11

**Rating:** 6
**Confidence:** 4
**Soundness:** 3 good
**Presentation:** 2 fair
**Contribution:** 3 good

**Summary:**

In this paper, the authors propose a method called amortized mixing coupling processes for clustering (AMCP). ACMP learns the relationship between samples and reference distribution with intra-cluster mixing (IntraCM); while assigning samples to clusters and generating new clusters with inter-cluster coupling (InterCC). The ACMP method is tested on synthetic and real-world datasets in the experiments.

**Questions:**

1. What are the contributions of this paper?
2. What is the assumption of the proposed clustering method? How is it different from existing amortized methods?
3. Is the objective function defined in Equation (11) guaranteed to converge?
4. In Table 1, does the time mean training time?


**Limitations:**

The authors did not address the limitations and potential negative social impact of their work.


**Strengths And Weaknesses:**

Strengths:

The authors propose a new clustering method called AMCP.

The method outperforms its competitors in terms of clustering performance and computational efficiency.

Weaknesses:

The contributions of the paper are not clear. The authors do not introduce the related work in detail, and it is not clear how the proposed method differs from the existing methods. How is the InterCC related other amortized clustering methods? Is the paper the first one that makes use of optimal transport for solving a clustering problem?

The authors might need to briefly introduce an optimal transport plan, entropic regularized Kantorovich relaxation, and how these concepts are related to a clustering problem. The assumption behind the proposed clustering method is also not apparent. Why is the relationship between the samples and reference distribution a proper way to define clusters?

I am not sure whether the proposed optimization procedure converges. It looks like in Equation (11), $b_{i,j}^{(k)}$ and $C_k$ are dependent. Therefore, dividing the optimization process into IntraCM and InterCC might not guarantee the convergence of the algorithm.

---

> ### Author Response · Authors · 2022-08-01
> **Response to Reviewer qZcB**
>
> Thanks very much for your valuable and comprehensive reviews!
>
> **Q1. The contributions of this paper.**
>
> Here we give the summarized contributions of our work as follow:
> - *Efficiency*. The proposed AMCP not only inherent the efficient cluster-wise learning manner, but also reduce the quadratic cost of traditional similarity weight computation with the aid of optimal transport mixing.
> - *Novelty*. To the best of our knowledge, AMCP is the first to introduce optimal transport for amortized clustering and effectively correlate the relationship between optimal transport and mixture models.
> - *Adaptivity*. Non-parametric learning manners exist not only in model parameter learning but also in cluster generation, which allows unsupervised parameter learning and non-handcrafted intervention.
> - *Flexibility*. Different from existing amortized clustering methods without changing the already-generated clusters, AMCP is able to dynamically adjust the previous generated clusters to avoid the accumulated errors with the aid of mixing and coupling manners.
> - *Effectiveness*. Extensive experiments conducted on both synthetic and real-world datasets demonstrate that AMCP can cluster data effectively and efficiently.
>
> We have added the above summarized contributions to Introduction part in the revised version.
>
> **Q2. How is it different from existing amortized methods?**
>
> Among existing amortized clustering models, ST-ACT inherits the feature of GMM which needs to manually set the number of clusters. DAC generalizes ST-ACT and enables to produce a varying number of clusters. The attention mechanism used in ST-ACT and DAC are much complex than the proposed optimal transport mixing reducing the quadratic cost of traditional similarity weight computation. The most similar work to our method is NCP and CHiGac. Both of them explain clustering from the perspective of generative model, which relies heavily on the selecting of anchor points and the processing ordering and often exhibits unstable properties. Different from previous work, AMCP connects the relationship of mixture model and optimal transport to describe the cluster's summary statistics well, and seamlessly combines mixing and coupling to achieve efficient amortized clustering.
>
> We have reorganized the related work, and clear how the proposed method differs from the existing work in the revised version. More detailed analyses are also given in revised Supplementary Materials (B.4).
>
> **Q3. Lack of introduction of optimal transport (OT). What is the assumption of the proposed clustering method?**
>
> Actually, we have given a detailed introduction of OT, entropic regularized Kantorovich relaxation, and relations to a clustering problem in original Supplementary Material (A and B). In the revised version, we have slightly reorganized the section 3.2.1 and give more introduction of OT and it’s connection to our method.
>
> In AMCP, the introduced IntraCM sufficiently exploit pairwise interactions between data points in intra-cluster level and benefit for the next inter-cluster coupling, which is able to sufficiently mine the hidden structure among data. Unlike the quadratic cost of the self attention mechanism used in ST-ACT and DAC, IntraCM uses a fixed number of references serving as queries, which is computational efficient, scalable to large dataset. We have cleared this point in section 3.2.1.
>
> **Q4. The converge guarantee of objective function.**
>
> By repeating the cluster generative process until there are no data points left, IntraCM and InterCC are processed alternatively, and the clusters are iteratively refined by deriving the explicit supervisory signal from the already formed clusters. This type of learning procedure is similar to the two-stage unsupervised clustering method, e.g., DeepCluster [1], which adopts pseudo labels from clusters. Although our AMCP does not contain a pseudo-label supervised learning process, IntraCM is performed on both already formed clusters and remaining unassigned data points, which can be regarded as learning process for pseudo labels. In this case, we can say AMCP is able to converge. We have cleared this point in the revised version. Furthermore, we provide the training and test performance vs. epoch for AMCP on synthetic data and CIFAR-10 in the Supplementary Material (C.2.3), which empirically proves the model convergence.
>
> [1]Caron M, et al. Deep clustering for unsupervised learning of visual features, ECCV. 2018
>
> **Q5. In Table 1, does the time mean training time?**
>
> The time listed in both Table 1 and 2 is average testing time on 5 runs since we want to show how quickly the model scales to the unseen test classes. We also provide training times in Supplementary Material (C.2.5).
>
> **Q6. The limitation of this work.**
>
> Limitations of this work are presented in the conclusion, such as the inapplicability to hierarchical data and the lack of interpretability and controllability.
>
>
> Finally, thanks again for your valuable comments.

---

### Author Response · Authors · 2022-08-01
**Response**

We would like to thank the reviewers for their helpful feedback and insightful comments and AC and SAC for their efforts in the review work. We answered the questions raised by each reviewer individually. The revised paper and corresponding supplementary material were also submitted. The modified parts are marked in blue font. Thanks again.

---

### Meta-Review · Area_Chair_ZcCd · 2022-08-23

**Recommendation:** Accept
**Confidence:** Certain

**Metareview:**

In this paper. the authors propose a novel amortized clustering method in which intra-cluster mixing and inter-cluster coupling are introduced. The optimal transport is used to learn the relationship between samles and reference distribution with intra-cluster mixing. The inter-cluster coupling assign samples to clusters and generates new clusters. The proposed method is novel in the sense that optimal transport is first introduced to amortized clustering, and its effectiveness is shown though synthetic and real datasets.
Many readers would be interested in this novel approach.

**Award:**

No

---

### Decision · Program_Chairs · 2022-09-14

Accept